# Rotate the ReLU to Sparsify Deep Networks Implicitly

**Nancy Nayak**                                                                           *ee17d408@smail.iitm.ac.in*
*Department of Electrical Engineering*
*Indian Institute of Technology Madras, India*

**Sheetal Kalyani**                                                                        *skalyani@ee.iitm.ac.in*
*Department of Electrical Engineering*
*Indian Institute of Technology Madras, India*

**Reviewed on OpenReview:** *https://openreview.net/forum?id=NzyOXmCPuZ*

## Abstract

Compact and energy-efficient models have become essential in this era when deep learning-based solutions are widely used for various real-life tasks. In this paper, we propose rotating the ReLU activation to give an additional degree of freedom in conjunction with the appropriate initialization of the rotation. This combination leads to implicit sparsification without the use of a regularizer. We show that this rotated ReLU (RReLU) activation improves the representation capability of the parameters/filters in the network and eliminates those parameters/filters that are not crucial for the task, giving rise to significant savings in memory and computation. While the state-of-the-art regularization-based Network-Slimming method achieves 32.33% saving in memory and 26.38% saving in computation with ResNet-164, RReLU achieves a saving of 35.92% in memory and 25.97% in the computation with a better accuracy. The savings in memory and computation further increase by 64.67% and 52.96%, respectively, with the introduction of $L_1$ regularization to the RReLU slopes. We note that the slopes of the rotated ReLU activations act as coarse feature extractors and can eliminate unnecessary features before retraining. Our studies indicate that features always choose to pass through a lesser number of filters. We demonstrate the results with popular datasets such as MNIST, CIFAR-10, CIFAR-100, SVHN, and Imagenet with different architectures, including Vision Transformers and EfficientNet. We also briefly study the impact of adversarial attacks on RReLU-based ResNets and observe that we get better adversarial accuracy for the architectures with RReLU than ReLU. We also demonstrate how this concept of rotation can be applied to the GELU and SiLU activation functions, commonly utilized in Transformer and EfficientNet architectures, respectively. The proposed method can be utilized by combining with other structural pruning methods resulting in better sparsity. For the GELU-based multi-layer perceptron (MLP) part of the Transformer, we obtain 2.6% improvement in accuracy with 6.32% saving in both memory and computation.

## 1 Introduction

Machine Learning has garnered significant attention in recent times for achieving superhuman-level performance in a wide range of problem domains, encompassing real-world applications and intricate tasks. This remarkable progress is primarily attributed to the utilization of Deep Neural Networks (DNNs) endowed with millions of parameters. In the realm of image classification, Convolutional Neural Networks (CNNs) such as AlexNet (Krizhevsky et al., 2012) with 60 million parameters and VGGNet (Simonyan & Zisserman, 2014) with 138 million parameters have set the benchmark. However, deep networks are prone to overfitting and encounter exploding and vanishing gradient problems. To address this issue, He et al. (2016a) introduce ResNet, a neural architecture that surpasses VGGNet in depth and can be extended to include thousands of layers while maintaining relatively lower computational complexity. Zagoruyko & Komodakis (2016) illustrate that achieving a marginal improvement in ResNet performance necessitates doubling the

number of layers. Consequently, they introduce WideResNet (WRN) as an alternative approach to enhance performance by widening the ResNet architectures instead of increasing their depth.

Increasing depth or width to improve accuracy also leads to storing a significantly higher number of parameters. For resource-constrained, energy-efficient green networks, the major concerns regarding the deployable network are (i) model size, (ii) run-time memory, and (iii) computation in terms of floating point operations (FLOPs). In order to reduce the complexity without degrading the performance, different compression techniques have been introduced - for example, low-rank approximation (Denton et al., 2014), quantization (Han et al., 2015), binarization (Courbariaux et al., 2015; Rastegari et al., 2016), and transfer learning (Kim et al., 2020). Numerous studies have been conducted (Han et al., 2015; Molchanov et al., 2016; Li et al., 2017; He et al., 2017; Liu et al., 2017) that put forth methodologies involving the pruning of connections, weights, and filters in deep neural networks. Typically, regularization techniques are employed to induce sparsity in weight matrices as a means of facilitating efficient pruning, as demonstrated in previous studies (Han et al., 2015; Wen et al., 2016; Liu et al., 2019). However, these techniques primarily target memory reduction without significantly impacting computational efficiency. To address the dual objectives of memory and computation savings, a group sparsity-based regularization approach has been introduced in the context of CNNs in various investigations conducted by Meier et al. (2008); Wang et al. (2017); Nayak et al. (2020); Zhou et al. (2016); Liu et al. (2015); Scardapane et al. (2017).

In Liu et al. (2017), $L_1$ regularization is imposed on the scaling factors of Batch Normalization (BN) layers, making it easy to implement without introducing any changes to the existing CNN architectures. By using the idea of the $L_1$ penalty on BN scaling parameters, MorphNet (Gordon et al., 2018) iteratively shrinks and expands a network, followed by retraining. It shrinks by a resource (FLOP or memory)-weighted sparsifying regularizer on activations and expands by a uniform multiplicative factor on all the layers. ThiNet is proposed by Luo et al. (2017) where the algorithm first trains a model, and then, by a greedy algorithm, it determines the filters that can be removed without affecting the performance. In Liu et al. (2019), the authors claim to perform better when the pruned networks are further retrained from scratch. Yang et al. (2020a) propose decomposed training that reduces the rank of the matrix/kernel via SVD training, achieving a higher reduction in computation load under the same accuracy. However, it requires separate regularizers for the orthogonality of the singular vectors and sparsity of the singular values.

Until now, researchers have used regularizers to impose sparsity, forcing the deeper/wider networks to use fewer filters without degrading the performance. However, **can this sparsity be achieved intrinsically without a regularizer?** Motivated by this question, in this work, we propose a novel activation function *Rotated ReLU (RReLU)*, where one of the two halves of a ReLU can rotate to have any slope, whereas the other half has a slope as zero. This slope of the rotating half is trained along with the other network parameters. The proposed method is different from dynamic ReLU (Chen et al., 2020) because, in dynamic ReLU, both halves have different slopes determined by a hyper-function dependent on all the input elements. In RReLU, as the slopes are trained along with the network, each of the filters in the convolution layer with a significant RReLU slope attains more representation power. During the training, some of the RReLU slopes corresponding to the excess filters go to very small values. Therefore, introducing rotation to ReLU has a two-fold advantage - (i) improves representation power corresponding to each significant filter and, therefore, (ii) encourages sparsity. This efficient sparsifying method allows pruning filters efficiently without using any regularizer. Note that researchers have come up with different variations of the activation ReLU (Nair & Hinton, 2010) such as leaky ReLU (Maas et al., 2013), PReLU (He et al., 2015), Exponential Linear Units (ELU) (Clevert et al., 2015), randomized ReLU (Xu et al., 2015) and randomized leaky ReLU (Srivastava et al., 2014) to improve the performance. Dynamic ReLU (Chen et al., 2020) is proposed to improve the representation power of deep networks. This activation function is characterized by piecewise linearity with varying slopes, and its parameters are determined by a hyper function over all input elements. Hendrycks & Gimpel (2016) propose Gaussian Error Linear Units (GELU), which is shown to improve the performance compared to ReLU and ELU. However, none of the above induces sparsity.

We discuss RReLU in detail in Sec. 2. To demonstrate the power of RReLU, we choose ResNet-based architectures, which are extensively used in diverse areas such as computer vision, natural language processing, security, healthcare, remote sensing, and wireless communication (Guo et al., 2020; Nguyen et al., 2021; Alrabeiah et al., 2020; Mañas et al., 2021; Shankar et al., 2021). However, irrespective of the architecture,

any model that has ReLU can be retrained by replacing the ReLUs with RReLUs. Further, we show that the use of rotation in activation is not only limited to ReLU but can be applied to other activations as well, like Gaussian Error Linear Unit (GELU) (Hendrycks & Gimpel, 2016) that are used in emerging transformer architectures. Researchers use GELU instead of ReLU for many applications to achieve state-of-the-art results using Transformers, including Natural Language Processing (Radford et al., 2018) and Vision Transformers (Dosovitskiy et al., 2020). By proposing rotated GELU (RGELU) in Vision Transformer, we not only show that sparsity is induced, but the top-1 validation accuracy improves, too, thus making it suitable across different architectures. This makes the RReLU, in our opinion, a potent tool for compressing neural networks. Our key contributions are as follows.

- We propose a Rotated ReLU activation along with an initialization scheme for the rotation. We show that this combination is a powerful tool to improve the representation capability of the filters, resulting in intrinsic sparsification. For example, the WRN-40-4 architecture with RReLU saves a memory of 63.37% and FLOP-count of 69.7% without any loss in performance.

- RReLU can work with other compression techniques. An example considered in our work is imposing $L_1$ regularization of batchnorm parameters. Using RReLU with the $L_1$ regularization of batchnorm scaling parameters achieves better sparsity and accuracy.

- RReLU provides better adversarial robustness when compared with ReLU in networks such as ResNet-56.

- We validate the results with extensive simulation [1] with various architectures such as fully connected neural network (FCNN), ResNet, WideResNet, and Transformer using a wide variety of datasets such as MNIST, CIFAR-10, CIFAR-100, SVHN, and large-scale image dataset like ILSVRC-2012 (ImageNet-1k).

- Finally, we show that the idea of rotation is not limited to ReLU and can be applied to activations like GELU used in most popular Transformer architectures.

## 2 Why and how to rotate the ReLU?

ReLU is a widely used activation function for introducing non-linearity in DNNs. The output of $l^{th}$ layer of a DNN of depth $L$ is given by $\mathbf{h}_{l+1} = \sigma(\mathcal{F}(\mathbf{h}_l; \mathbf{W}_l))$, where $\mathcal{F}$ is either the weights multiplication operation for FCNN or the convolution operation for a CNN followed by batchnorm layer, $\mathbf{W}_l$ are the trainable weights or filters respectively of layer $l$, and $\sigma$ is the non-linearity or the activation function. The ReLU activation (Glorot et al., 2011) $\sigma(x) = \max(0, x)$ is a composite of piecewise linear approximation, hence can approximate a wide range of functions (Huang, 2020; Hanin, 2019; Cybenko, 1989; Barron, 1993). The non-zero-slope part of ReLU has a fixed slope of 1. In this paper, we increase the degrees of freedom of ReLU

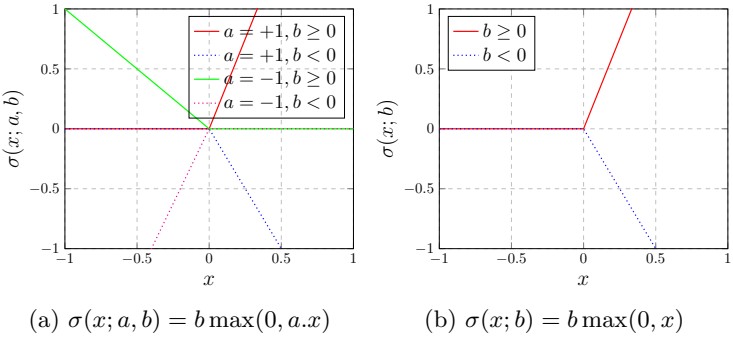

(a) $\sigma(x; a, b) = b \max(0, a.x)$    (b) $\sigma(x; b) = b \max(0, x)$

Figure 1: Rotated ReLU activation

---

[1]Code: `https://github.com/nancy-nayak/RotatedReLU_TMLR`

by rotating the part of the activation where the slope is one, with a non-zero slope, thereby increasing the representative power of ReLU. The proposed RReLU activation $\sigma(x; a, b) = b \max(0, ax)$ has two trainable parameters $a \in \{+1, -1\}$ and $b \in \mathbb{R}$. RReLU outputs zero for either the positive part or the negative part of its domain, and for the remaining part of the domain, it produces a linear output. The slope of this linear output depends on the values of $(a, b)$. With these values of $a$ and $b$, various possible RReLUs can be realized. Fig. 1a shows all possible cases of RReLU activations with different $(a, b)$.

Employing an RReLU with shared $(a, b)$ values for all features across all layers is quite similar to using regular ReLU in the architecture, as the roles of $a$ and $b$ are compensated by the network weights/filters. Therefore, to fully exploit the power of rotation, instead of using a single $(a, b)$ across all features in all layers, $(a_l^{\{i\}}, b_l^{\{i\}})$ is used for $i^{th}$ feature $x_l^{\{i\}}$. So, the output of the $l^{th}$ layer is given by

$$\mathbf{h}_{l+1} = \sigma(\mathbf{x}_l; \mathbf{a}_l, \mathbf{b}_l) = \mathbf{b}_l \max(0, \mathbf{a}_l . \mathbf{x}_l) \tag{1}$$

where the feature input to the RReLU is $\mathbf{x}_l = \mathcal{F}(\mathbf{h}_l; \mathbf{W}_l)$. As $\mathbf{x}_l$ is the output from the linear/convolution layer, it is clear from equation 1 that any numeric value $\mathbf{a}_l$ can take can be adjusted using the weights/filters itself. Therefore, only two different types of RReLU corresponding to $b \geq 0$ and $b < 0$ are sufficient for the proposed technique and are shown in Fig.1b. With these RReLU activations, the output of the $l^{th}$ layer is denoted as

$$\mathbf{h}_{l+1} = \sigma(\mathbf{x}_l; \mathbf{b}_l) = \mathbf{b}_l \max(0, \mathbf{x}_l). \tag{2}$$

When the input $\mathbf{h}_l$ is fed to the $l^{th}$ layer within a residual unit, the output of the unit with a ReLU activation can be represented by

$$\mathbf{h}_{l+2} = \max(0, \mathbf{h}_l + \boldsymbol{\gamma}_{l+1} \text{Conv}(\max(0, \boldsymbol{\gamma}_l \text{Conv}(\mathbf{h}_l; \mathbf{W}_l) + \boldsymbol{\beta}_l); \mathbf{W}_{l+1}) + \boldsymbol{\beta}_{l+1}), \tag{3}$$

where $\boldsymbol{\gamma}_l$ and $\boldsymbol{\beta}_l$ are the batchnorm scaling and shifting parameters, respectively. Incorporating $\mathbf{b}_l$ in equation 3 increases the capacity of the piecewise linear approximation to represent the underlying function with more linear segments. The output of the residual unit with an RReLU activation is given by

$$\mathbf{h}_{l+2} = \mathbf{b}_{l+1} \max(0, \mathbf{h}_l + \boldsymbol{\gamma}_{l+1} \text{Conv}(\mathbf{b}_l \max(0, \boldsymbol{\gamma}_l \text{Conv}(\mathbf{h}_l; \mathbf{W}_l) + \boldsymbol{\beta}_l); \mathbf{W}_{l+1}) + \boldsymbol{\beta}_{l+1}). \tag{4}$$

**Sparsity and RReLU:** Due to the freedom in choosing the value of the slope, the architecture with RReLU needs lesser trainable weights to achieve a similar performance as ReLU. The degree of rotation of the non-zero-slope segment of RReLU determines the degree to which the feature $\mathbf{x}_l$ is transmitted through the $l^{th}$ layer. Once fully trained, if the value of $b_l^{\{i\}}$ for $i^{th}$ feature is comparatively less, then it implies that the convoluted feature $x_l^{\{i\}}$ is not essential for the task and can be ignored keeping the performance intact. It has been observed that with the introduction of RReLU, for deeper/wider architectures, many of the $\mathbf{b}_l$ go to zero, resulting in a significant amount of structural sparsity in the model. Note that the value of $h_{l+1}^{\{i\}}$ is insignificant when $b_l^{\{i\}}$ is approximately equal to zero, and this is logically consistent as $\mathbf{x}_l$ is batch normalized before passing it via RReLU, and

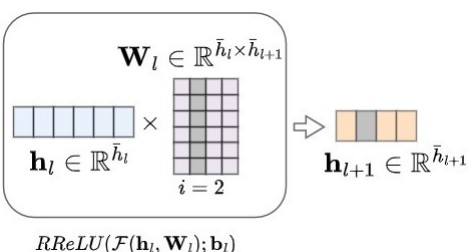

$$RReLU(\mathcal{F}(\mathbf{h}_l, \mathbf{W}_l); \mathbf{b}_l)$$

Figure 2: RReLU in FCNN. The figure is better visible in color. $\mathcal{F}(.)$ represents linear operations with learnable parameters $\mathbf{W}_l$. The RReLU slopes $\mathbf{b}_l$, too, are the learnable parameters.

therefore $\mathbf{x}_l$ can take only a bounded value. We do not use magnitude-based pruning on weights/kernels; instead, RReLU inherently learns sparsity, and we prune weights/filters based on the corresponding RReLU slope values. Now, we describe how rotation helps to achieve lower complexity with lesser computation. We discuss the results using FCNN and ResNet architectures, where the primary operations are matrix multiplication and convolution with skip connection, respectively.

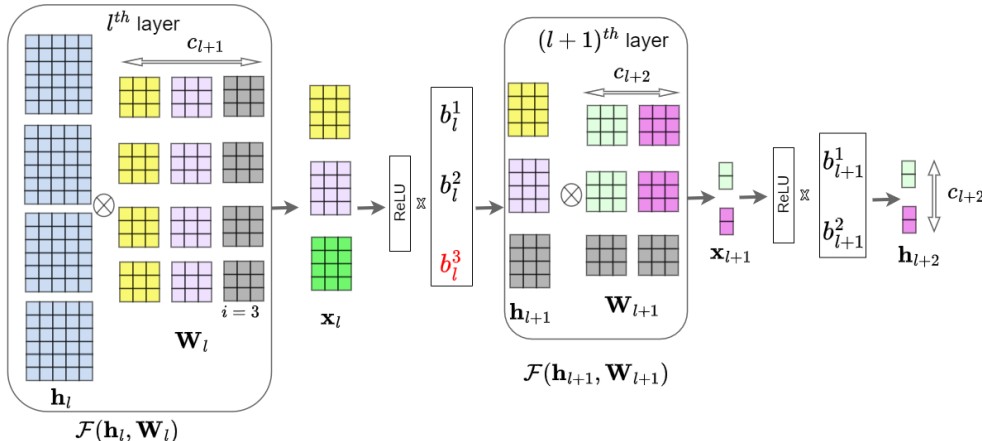

Figure 3: RReLU in CNN. The figure is better visible in color. $\mathcal{F}(.)$ represents convolution operation with learnable parameters $\mathbf{W}_l$. The RReLU slopes $\mathbf{b}_l$, too, are learnable.

**RReLU in FCNN:** Let the input and output for the $l^{th}$ layer be $\mathbf{h}_l$ and $\mathbf{h}_{l+1}$, respectively and let the number of elements in $\mathbf{h}_{l+1}$ be $\bar{h}_{l+1}$. The RReLU activation with trainable slope $\mathbf{b}_l \in \mathbb{R}^{\bar{h}_{l+1}}$ is applied on $\mathcal{F}(\mathbf{h}_l, \mathbf{W}_l)$. After training, element $b_l^{\{i\}}$ is compared against a threshold $\zeta$ and made equal to zero if $b_l^{\{i\}}$ is less than $\zeta$, indicating that the $i^{th}$ column of the weight matrix $\mathbf{W}_l$ can be ignored. The threshold $\zeta$ is set by cross-validation. This idea is pictorially illustrated with an example in Fig. 2. Here, $\mathbf{F}(.)$ is matrix multiplication and denoted by $\times$. In the example in Fig. 2, after the training, the second element of $\mathbf{b}_l$ converges to zero, i.e., $b_l^{\{2\}} \to 0$. So the $2^{nd}$ entry in $\mathbf{h}_l$ is negligible as the RReLU does not allow the second element of $\mathbf{x}_l$ to pass; clearly, the $2^{nd}$ column of $\mathbf{W}_l$ can be ignored.

**RReLU in CNN:** Let the number of channels out from the convolution operation at the $l^{th}$ layer be $c_{l+1}$. The RReLU activation with trainable slope $\mathbf{b}_l \in \mathbb{R}^{c_{l+1}}$ is applied on $\mathcal{F}(\mathbf{h}_l, \mathbf{W}_l)$. After training, $b_l^{\{i\}}$ is compared against a threshold $\zeta$, and if $b_l^{\{i\}} < \zeta$ then the $i^{th}$ feature at the $l^{th}$ layer can be ignored. Therefore, at the $l^{th}$ layer, $i^{th}$ filter out of a set of $c_{l+1}$ filters can be ignored and at $(l+1)^{th}$ layer, $i^{th}$ sub-filter from each of $c_{l+2}$ filters can be ignored. This is explained pictorially with an example in Fig. 3 where $\otimes$ denotes the convolution operation. At the $l^{th}$ layer, four 2D features $\mathbf{h}_l$ are convolved with three set of filters denoted by $\mathbf{W}_l$ with four sub-filters each, resulting in $\mathbf{h}_{l+1}$. The first 2D feature (yellow) of $\mathbf{h}_{l+1}$ is calculated by convolving each of the four 2D features in $\mathbf{h}_l$ with corresponding sub-filters of the first filter (yellow) and adding them. The same occurs at layer $l+1$. Let, after the training, the slope $b_l^{\{3\}} \to 0$, then $3^{rd}$ feature in $\mathbf{h}_{l+1}$ is also close to zero. In this case, the $3^{rd}$ filter of $\mathbf{W}_l$ and the $3^{rd}$ sub-filter of every filter in $\mathbf{W}_{l+1}$ can be ignored (grey).

## 2.1 Enhanced Filter Representation with RReLU and Batch Normalization Scaling Parameters

Batchnorm scaling and shifting parameters $\boldsymbol{\gamma}_l$ and $\boldsymbol{\beta}_l$, respectively, in equation 3 are used to scale and shift the batch normalized features to restore the representation power of the network after normalization (Ioffe & Szegedy, 2015). The parameters are learned along with the original model parameters. Liu et al. (2017) propose a technique, namely network-slimming for structural pruning, which involves the utilization of the $L_1$ norm on the batch normalization scaling parameters $\boldsymbol{\gamma}_l, \forall l \in L$. This approach encourages the $\boldsymbol{\gamma}_l$ parameters to approach zero, allowing for the corresponding filters to be discarded. Our proposed method provides increased flexibility to the $\boldsymbol{\gamma}_l$ by introducing slopes $\mathbf{b}_l$ as shown in equation 4, which enhances the representation power corresponding to every filter in all the convolution layers. In the context of this study, an increase in filter representation power signifies the achievement of comparable accuracy while utilizing a reduced number of filters. *As shown in equation 4, when some of the slopes ($\mathbf{b}_l$) go to zero, then corresponding BN parameters ($\boldsymbol{\gamma}_l$) go to zeros too. In case some elements of $\mathbf{b}_l$ and the corresponding elements of $\boldsymbol{\gamma}_l$ do not go to zero, then the elements of $\mathbf{b}_l$ give more flexibility to batchnorm parameters*

*resulting in more representation power to the filters, and as a result, many filters become unnecessary.* The degree of sparsity is further enhanced through the application of $L_1$ regularization to the RReLU slopes. To delve deeper into this, we need a more thorough analysis. Liu et al. (2017) minimizes the $L_1$ norm on the $\boldsymbol{\gamma}_l$ given by $\sum_{l=1}^{L} \sum_{i=1}^{N} \gamma_l^{\{i\}}$ that will force each element $\gamma_l^{\{i\}}$ to take lower values and therefore the network may lose its representation power. On the other hand, while minimizing $L_1$ norm on RReLU slopes, although every element of $\mathbf{b}_l$ is compelled to adopt smaller values, the term $b_l^{\{i\}} \gamma_l^{\{i\}}$ remains unconstrained because the regularization is not directly applied to the scaling parameters, $\boldsymbol{\gamma}_l$. Thus, RReLU possesses the potential to augment the representational capabilities of every filter within the convolutional layers to a greater extent than ReLU in scenarios both with and without the application of regularization. Moreover, we have substantiated this assertion through empirical evidence, as detailed in Section 5.5.

## 2.2 Compatibility of RReLU with other algorithms Featuring ReLU

RReLU is applicable to other architectures as well, where ReLU is used as an activation function. The output of the residual unit of PreAct-ResNet (He et al., 2016b) is given by

$$\mathbf{z}_{l+2} = \mathbf{z}_l + \text{Conv}(\max(0, \boldsymbol{\gamma}_{l+1}\text{Conv}(\max(0, \boldsymbol{\gamma}_l\mathbf{z}_l + \boldsymbol{\beta}_l); \mathbf{W}_l) + \boldsymbol{\beta}_{l+1}); \mathbf{W}_{l+1}). \tag{5}$$

When ReLU is replaced by RReLU, the same is represented by

$$\mathbf{z}_{l+2} = \mathbf{z}_l + \text{Conv}(\mathbf{b}_{l+1}\max(0, \boldsymbol{\gamma}_{l+1}\text{Conv}(\mathbf{b}_l\max(0, \boldsymbol{\gamma}_l\mathbf{z}_l + \boldsymbol{\beta}_l); \mathbf{W}_l) + \boldsymbol{\beta}_{l+1}); \mathbf{W}_{l+1}). \tag{6}$$

Similarly, the proposed RReLU activation can be used along with other compression techniques as well, such as ThiNet (Luo et al., 2017) and MorphNet (Gordon et al., 2018). The main idea of ThiNet (Luo et al., 2017) is to approximate the output of a layer by using a subset of the filters in that layer to remove the rest of the filters safely. For filter selection, they use a data-driven greedy algorithm. One can use RReLU instead of ReLU in the ThiNet architecture to study if one can prune more filters with RReLU, which could be an interesting future study. For shrinking a network, MorphNet (Gordon et al., 2018) considers a sparsifying regularizer $\mathcal{G}(.)$ based on discounting the use of batchnorm parameters,

$$\mathcal{G}(.) = \sum_{l=1}^{L} \left( C \sum_{i=0}^{I_l-1} |\gamma_{l-1,i}| \sum_{j=0}^{O_l-1} B_{l,j} + C \sum_{i=0}^{I_l-1} A_{l,i} \sum_{j=0}^{O_l-1} |\gamma_{l,j}| \right). \tag{7}$$

Here $I_l$ and $O_l$ denotes the number of input and output channel, respectively, for the $l^{th}$ layer; $A_{L,i}(B_{L,j})$ is an indicator function which equals one if the $i^{th}$ input ($j^{th}$ output) of layer $L$ is not zeroed out. The same can be done with RReLU in the architecture as well, where the same regularizer $\mathcal{G}(.)$ can be applied. Note that, here RReLU slopes provides better representation power of every filter in the architecture.

Yang et al. (2020a) propose singular value decomposition (SVD) training to explicitly achieve low-rank DNNs. The authors use sparsity-inducing regularizers on the singular value vector $\mathbf{s}$ of each layer, such that the $L_1$ regularization on $\mathbf{s}$ can be represented as $L^1(\mathbf{s}) = \sum_i s_i$. Therefore, all the singular values $s_i, \forall i$ shrink simultaneously to take lower values. To mitigate the proportional scaling associated with the $L_1$ regularization, the authors uses Hoyer regularizer (Hoyer, 2004) defined as $L^H(\mathbf{s}) = \|\mathbf{s}\|_1/\|\mathbf{s}\|_2 = \sum_i |s_i|/\sqrt{s_i^2}$. However, one can replace the ReLU activations with the proposed RReLU activation while still conducting SVD training to get another degree of freedom as the elements of $\mathbf{b}_l$ can take any values. This will provide higher representation power to every filter[2] which may increase the sparsity even more.

A channel exploration method has been recently proposed in Hou et al. (2022) which repeatedly prunes and regrows the channels throughout the training process. Channel pruning selects the most representative columns from $\mathbf{W}_l$ that can reconstruct the original channel matrix with minimal error, therefore formulated as a column subset selection (CSS) problem. The regrowing stage helps to dynamically re-allocating the number of channels across all the layers under a global channel sparsity constraint. CHEX is shown to have excellent pruning capability, and RReLU combined with CHEX can further improve the sparsity. Therefore, the proposed RReLU need not replace the current sparsifying techniques but can also complement them further to enhance sparsity.

---

[2]In this work, every filter is represented as a combination of orthogonal singular vectors and singular values.

## 2.3 Applicability of the idea of rotation to GELU activations

Till now, we have discussed how RReLU can be applied to replace ReLU in sparsification techniques. Now, we elucidate how the concept of rotation extends to GELU (Hendrycks & Gimpel, 2016), which is defined by the equation:

$$\sigma_{GELU}(x) = xP(X \leq x) = x\phi(x) = x.\frac{1}{2}\left[1 + \text{erf}(\frac{x}{\sqrt{2}})\right]. \tag{8}$$

It is well-established that Transformers exhibit improved performance when employing GELU activations (Radford et al., 2018). To introduce varying slopes in the GELU activation, we propose the rotation of the GELU activation, denoted as *RGELU*, as follows:

$$\mathbf{h}_{l+1} = \sigma_{RGELU}(\mathbf{x}_l; \mathbf{b}_l) = \mathbf{b}_l\mathbf{x}_l.\frac{1}{2}\left[1 + \text{erf}(\frac{\mathbf{x}_l}{\sqrt{2}})\right] \tag{9}$$

Therefore, any architecture that uses activations similar to ReLU can use the rotated version of the activation of variable slope. For example, SiLU used in EfficientNets can be replaced by rotated SiLU (RSiLU) using the above idea of rotation. In Fig. 4, we show how RGELU can have any slope compared to GELU, which is fixed at $b_l = 1$. In Sec. 5, we provide results with Vision Transformers and EfficientNet on the ImageNet dataset that prove the intrinsic sparsification properties of RGELU and RSiLU, respectively.

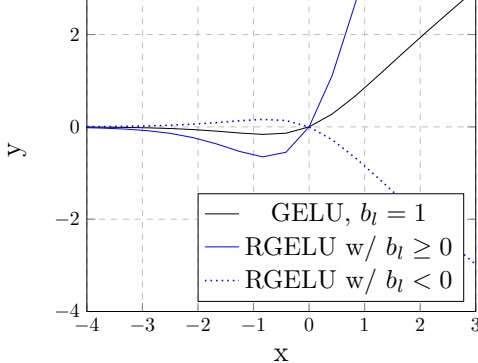

Figure 4: Rotated GELU activation $\sigma_{GELU}(\mathbf{x}_l; \mathbf{b}_l)$

## 3 Saving memory and FLOPs using RReLU

For an FCNN, considering $\mathbf{W}_l \in \mathbb{R}^{\bar{h}_{l+1} \times \bar{h}_l}$, the total number of parameters at the $l^{th}$ layer is given by $\bar{h}_l\bar{h}_{l+1}$. During the forward pass, the number of multiplications and additions is given by $\bar{h}_{l-1}\bar{h}_l$ and $(\bar{h}_{l-1}-1)\bar{h}_l$ respectively; therefore, a total of $\approx 2\bar{h}_{l-1}\bar{h}_l$ FLOPs are needed. If $n$ values of $\mathbf{b}_l$ are zero, then $n$ columns of $\mathbf{W}_l$ can be ignored. So, the memory saving at the $l^{th}$ and the $(l+1)^{th}$ layer are $\bar{h}_{l-1}n$ and $n\bar{h}_{l+1}$ respectively. The savings in FLOPs for the $l^{th}$ and the $(l+1)^{th}$ layers are approximately $2\bar{h}_{l-1}n$ and $2n\bar{h}_{l+1}$.

The savings are even more for the CNN-based ResNets. The memory and FLOPs for 1D-CNN and 2D-CNN were given in Vikas et al. (2021) and Shankar et al. (2021), respectively. Considering $L$ layers in a ResNet architecture, the trainable parameters are given by $\phi = \{\mathbf{W}_1, \ldots, \mathbf{W}_L\}$ where $\mathbf{W}_l \in \mathbb{R}^{c_{l+1} \times c_l \times k \times k}$ is the filter for the $l^{th}$ layer of a 2D CNN. Here, $c_l$ and $c_{l+1}$ represent the number of input and output channels at the $l^{th}$ layer, respectively, and $k$ is the dimension of the filter. The input and output for the $l^{th}$ layer are $\mathbf{h}_l \in \mathbb{R}^{c_l \times \bar{h}_l^w \times \bar{h}_l^h}$ and $\mathbf{h}_{l+1} \in \mathbb{R}^{c_{l+1} \times \bar{h}_{l+1}^w \times \bar{h}_{l+1}^h}$ respectively. Here, $(\bar{h}_l^w, \bar{h}_l^h)$ and $(h_{l+1}^w, h_{l+1}^h)$ are spatial dimensions (width, height) of the input and the output respectively. The total number of multiplication for the $l^{th}$ layer is $c_l k^2 \bar{h}_{l+1}^w \bar{h}_{l+1}^h c_{l+1}$ and the total number of addition for the $l^{th}$ layer is $(c_l - 1)(k^2 - 1) \times \bar{h}_{l+1}^w \bar{h}_{l+1}^h c_{l+1}$. The total count of FLOPs for the $l^{th}$ layer of a real-valued 2D CNN is the summation of the number of multiplication and addition that is roughly twice the number of multiplication given by $2c_l k^2 \bar{h}_{l+1}^w \bar{h}_{l+1}^h c_{l+1}$. Because of the addition of residue in the ResNet structure, if there is a residual connection at the $l^{th}$ layer, an extra of $c_{l+1} \bar{h}_{l+1}^w \bar{h}_{l+1}^h$ additions are required. At $l^{th}$ convolution layer, if the input to RReLU at the $l^{th}$ layer has $c_{out}^l$ channels and $n$ entries of $\mathbf{b}_l$ are insignificant, then only $(c_{l+1} - n)$ channels remain significant. This needs to save only $(c_{l+1} - n)c_l k^2$ parameters for the $l^{th}$ layer. The computation for these reduced $(c_{l+1} - n)$ filters at the $l^{th}$ layer is only $2c_l k^2 \bar{h}_{l+1}^w \bar{h}_{l+1}^h (c_{l+1} - n)$ and the same at $(l+1)^{th}$ layer is given by $2(c_{l+1} - n)k^2 \bar{h}_{l+2}^w \bar{h}_{l+2}^h c_{l+2}$.

# 4  RReLU against Adversarial attacks

Deep neural networks are susceptible to adversarial attacks, wherein a seemingly minor malicious alteration in the input can lead to the network misclassifying the data point. Numerous techniques have been investigated with the aim of fortifying neural networks against such vulnerabilities, leading to the concept of adversarial robustness in the network (Cisse et al., 2017; Tramèr et al., 2017). We briefly investigate the adversarial robustness characteristics of networks with RReLU activation. Considering a function $f : \mathcal{X} \to \mathbb{R}^C$ as a neural network classifier with $C$ classes, (Cohen et al., 2019; Hein & Andriushchenko, 2017; Salman et al., 2019) showed that Lipschitzness of $f$ is closely related to its robustness. Yang et al. (2020b) demonstrated that a network could achieve better adversarial robustness by imposing an upper bound/constraint on the network's local Lipschitz constant (LLC). A smaller LLC indicates better adversarial robustness (Yang et al., 2020b; Tholeti & Kalyani, 2022). Note that Lipschitzness is a function of the gradients of the function $f$, which in turn is a function of the slopes of the RReLU. A function $f : \mathbb{R}^m \to \mathbb{R}^n$ is said to be Lipschitz continuous if $\forall \mathbf{x}, \mathbf{y} \in \mathbb{R}^m$,

$$|f(\mathbf{x}) - f(\mathbf{y})| \leq L_p ||\mathbf{x} - \mathbf{y}||_q \tag{10}$$

where

$$L_p = \sup\{||\nabla f(x)||_q : x \in \mathbb{R}^m\} \tag{11}$$

is the Lipschitz Constant (LC), $\nabla f(x)$ is gradient of function $f(x)$, $1/p + 1/q = 1$, $1 \leq p$ and $q \leq \infty$ (Latorre et al., 2020; Paulavičius & Žilinskas, 2006). Motivated by the fact that many of the RReLU slopes tend to take smaller values than one, we believed that the local Lipschitz constant for networks with RReLU should have lower values than the local Lipschitz constant for networks with ReLU. Therefore, the local smoothness should be more for architectures with RReLU. Yang et al. (2020b) also demonstrate that the usual robust training methods such as adversarial training (Madry et al., 2017), robust self-training (Raghunathan et al., 2020), and TRADES (Zhai et al., 2019) impose a high degree of local smoothness and therefore are most robust. However, these three methods have a large gap between training and test accuracies as well as adversarial training and test accuracies, leading to poor generalization of the robust methods. As we do not perform any adversarial training, the networks' generalization capability with RReLU is unaffected. Here, we study the adversarial robustness of RReLU by computing its LLC as it involves the gradient of the function $f(x)$ given by $\nabla f(x)$, which in turn is a function of RReLU slopes $\mathbf{b}_l$. Based on the results in subSec. 5.10, it seems that the architectures with RReLU are inherently more robust than those with ReLU.

# 5  Results and discussion

In order to demonstrate the effectiveness of our proposed method, we have conducted extensive investigations using various architectures, including a fully connected dense network, ResNet architectures (He et al., 2016a;b) with depths of 20, 56 layers, and WideResNet (Zagoruyko & Komodakis, 2016) with depths of 40 and 16, and a widening factor of 4. Inspired by He et al. (2016b), we have used improved ResNet architectures to efficiently train deeper ResNets (i.e., with depth 110 and 164). This architecture has identity mapping between two ResNet units where the convolution layer follows the ReLU layer, and we consider this as the baseline. As the activation is before the convolution layer, we name them ResNet-110-pre and ResNet-164-pre for depth 110 and 164, respectively.

We tested our method on various standard datasets such as MNIST, SVHN, CIFAR-10, and CIFAR-100. To evaluate our method, we compared its performance to the baseline performances of ResNet and WideResNets[3]. Towards the end, to show the efficacy of the proposed method and to strengthen our claims, we have demonstrated the results on the Imagenet dataset with both WideResNet and Transformer architectures (Vaswani et al., 2017). The experiments with Transformer architecture with the Imagenet dataset are conducted using an NVIDIA-A100 GPU, while the rest of them are conducted on a single NVIDIA-GeForce 2080 Ti GPU.

Even though the number of trainable parameters increases slightly in the architectures with RReLU, introducing RReLU can discard a significant amount of weights/filters. As the optimization process is sensitive

---

[3]as reported by Idelbayev; Kuen; Liu

| Dataset | MNIST | CIFAR-10 | | | | | CIFAR-100 | | | | | SVHN |
|---|---|---|---|---|---|---|---|---|---|---|---|---|
| Architecture | FCNN | ResNet-20 | ResNet-56 | ResNet-110-pre | ResNet-164-pre | WRN-40-4 | ResNet-20 | ResNet-56 | ResNet-110-pre | ResNet-164-pre | WRN-40-4 | WRN-16-4 |
| Acc ReLU (standard) | – | 91.25 | 93.03 | 93.63 | 94.58 | 95.47 | 68.20 | 69.99 | 74.84 | 75.67 | 78.82 | 97.01 |
| Acc ReLU (more training) | 98.33 | **93.12** | **94.45** | **95.33** | **95.51** | **96.18** | 67.27 | **72.91** | **75.23** | **78.31** | **79.75** | 96.75 |
| #Params ReLU | 0.10 | 0.27 | 0.85 | 1.7 | 1.7 | 8.9 | 0.27 | 0.85 | 1.7 | 1.7 | 8.9 | 2.78 |
| #FLOPs ReLU | 0.40 | 81 | 251 | 505 | 489 | 2605 | 81 | 251 | 505 | 489 | 2605 | 944 |
| Filters pruned (%) | 4.8 | 3.86 | 8.78 | 6.05 | 45.34 | 43.36 | 0.0 | 10.32 | 1.88 | 22.63 | 17.99 | 20.88 |
| Acc RReLU (post-pruning) | 98.24 | 92.86 | 94.11 | 95.11 | 95.10 | 96.01 | 67.69 | 71.40 | 74.87 | 77.85 | 80.19 | 96.77 |
| #Params RReLU | **0.09** | **0.25** | **0.78** | **1.59** | **0.85** | **3.26** | **0.27** | **0.798** | **1.69** | **1.48** | **7.23** | **1.86** |
| #FLOPs RReLU | **0.38** | **78** | **188** | **454** | **309** | **1245** | **81** | **175** | **487** | **409** | **1841** | **675** |

Table 1: Performance of RReLU in terms of accuracy, number of trainable parameters, and computation power (in FLOPs). Architectures with both ReLU and RReLU are trained from scratch without using any regularizer. The number of parameters and FLOPs are in Millions (Mn).

towards $\mathbf{b}_l$, the convergence is comparatively slow and, therefore, trained for more time. When some elements of the trainable $\mathbf{b}_l$ take very small values, the resulting network act as a sparse model. Research conducted by Kuznedelev et al. (2023) demonstrates that sparse networks exhibit a slower convergence rate and derive greater advantages from prolonged training sessions. Hence, while conventional models described in the literature undergo training for 200 epochs, models utilizing RReLU are trained for an extended duration of 1200 epochs. For a fair comparison, ReLU models are also trained for 1200 epochs, and the corresponding accuracy values are listed in the $2^{nd}$ row of Table 1. Please note that the models do not show a notion of overfitting as the validation accuracy keeps increasing till 1200 epochs. The last row of Table 1 represents the accuracy of networks when trained with RReLU. We prune the RReLU slopes using cross-validation such that the accuracy does not degrade after pruning. Comparing the accuracy of RReLU (in the last row) and ReLU (in the second row), one can observe that with RReLU, one can achieve an accuracy very close to ReLU even though we offer significant saving in memory and computation, which is evident by comparing #Params and #FLOPs for ReLU and RReLU.

Note that after 1200 epochs of training, the slight gap between the accuracies of ReLU and RReLU is because of slower convergence and not because of sparsification/compression. To empirically prove this, we trained ResNet20 and ResNet56 with RReLU for even higher (2000) training epochs; we achieved an accuracy of 93.25 and 94.14, which is comparable to the accuracy of 93.12 and 94.45 of respective architectures with ReLU. The WideResNet architecture WRN-40-4 for CIFAR10 and CIFAR100 datasets are trained for 400 epochs, and WRN-16-4 for the SVHN dataset is trained for 1000 epochs. All the architectures are trained using a Cosine Annealing (CA) learning rate (lr) scheduler.

## 5.1 More training improves the validation accuracy of the networks

Here, we make an interesting observation - when the networks are trained for a higher duration, the performance improves significantly for most of them, as highlighted in the second row in Table. 1. This observation is consistent with the findings of Nakkiran et al. (2021), which suggest that increasing the duration of training can be beneficial. The distribution of batchnorm parameters ($\gamma_l$) changes too from 200 epochs to 1200 epochs. Without employing any regularizer, an extended training duration compels a greater proportion of the elements in $\gamma_l$ to approach zero. As illustrated in Figure 5a, following 200 training epochs, there are approximately 50 elements in $\gamma_l$ in close proximity to zero. However, with prolonged training, this count rises to approximately 270, as depicted in Figure 5b. Hence, the augmentation of training duration facilitates a more pronounced adjustment of the $\gamma_l$ values, leading to improved representation. When employing RReLU in the architecture, the count of elements of $\gamma_l$ in proximity to zero further escalates, reaching an approximate value of 400, as depicted in Figure 5c. This observation suggests that a network equipped with RReLU attains greater flexibility, yielding a comparable representation capability with fewer filters compared to a basic ReLU-based ResNet architecture.

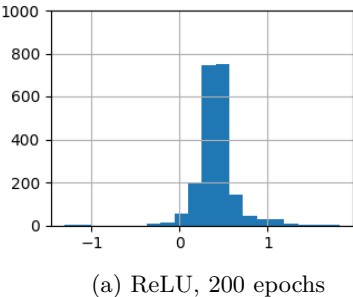 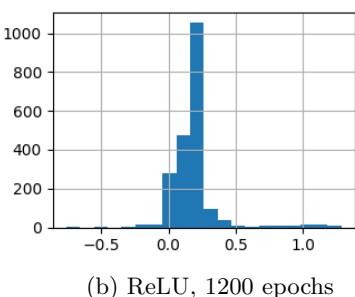 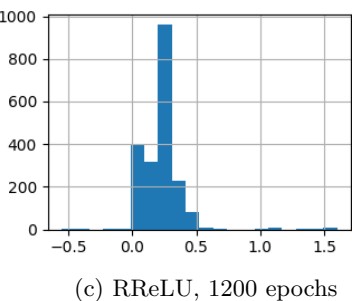

| (a) ReLU, 200 epochs | (b) ReLU, 1200 epochs | (c) RReLU, 1200 epochs |

Figure 5: Subfig. (a)-(c) are the distribution of batchnorm parameters $\boldsymbol{\gamma}_l$ after the training when the architecture considered is ResNet56. Subfig. (a)-(b) shows the effect of more training on $\boldsymbol{\gamma}_l$. Subfig. (c) shows the distribution of $\boldsymbol{\gamma}_l$ when ResNet56 is trained with RReLU for 1200 epochs.

## 5.2 Initialization of the network parameters

In this experimental setup, the weights/filters are initialized using the Kaiming He initialization method (He et al., 2015). We found that the right initialization of the RReLU slopes ($\mathbf{b}_l$) is crucial for effectively training the proposed method. When slopes are initialized randomly with a zero mean Gaussian distribution, many of the slopes $\mathbf{b}_l$ are initialized with very small values and converge to values close to zero very soon, and therefore, the whole network converges to a sub-optimal solution. Therefore, it is imperative to implement an effective initialization scheme for the slopes to ensure that the network has an adequate convergence window before certain slopes approach zero. Therefore, the slopes ($\mathbf{b}_l$ for all $l \in L$) are initialized with a truncated Gaussian Mixture Model (GMM) with a mean of $\{+1, -1\}$ and a variance of 3, which is depicted in Fig. 6. The utilization of the GMM-type initialization for $\mathbf{b}_l$ provides the slopes with a more extended duration for convergence, given that they are ini-

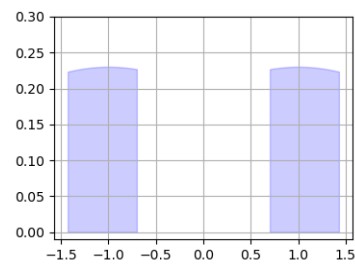

Figure 6: Initial Distribution of RReLU slopes ($\mathbf{b}_l$)

tialized with substantial values. After training, the RReLU slopes converge to a distribution where many elements of $\mathbf{b}_l$ tend towards zero, as can be observed in Fig. 7 across different architectures and datasets.

## 5.3 Distribution of RReLU slopes $\mathbf{b}_l$

Fig. 7 shows the converged RReLU slopes ($\mathbf{b}_l$) after the training for different architectures when trained with CIFAR10 and CIFAR100 datasets. A comparison of Fig. 7c and Fig. 7g reveals that for a relatively more challenging task, such as CIFAR100 classification using the ResNet-110-pre architecture, a smaller number of filters can be pruned using RReLU compared to a relatively easier task like CIFAR10 classification. Furthermore, the degree of pruning is more pronounced for a deeper architecture such as ResNet-164-pre in Fig. 7d than ResNet-110-pre in Fig. 7c. Readers are encouraged to check the distribution of slopes $\mathbf{b}_l$ for WideResNets in Fig. 15 in the Appendix, which shows that many filters are made redundant by the use of RReLU. Fig. 16 in the Appendix shows the distribution of BN parameters $\boldsymbol{\gamma}_l$ when different architectures are trained with CIFAR10 and CIFAR100 datasets. In the latter part of the paper, we discuss the potential range of values, both $\boldsymbol{\gamma}_l$ and $\mathbf{b}_l$ can take in Fig. 10.

## 5.4 Discussion on intrinsic sparsity property of RReLU

Table 1 presents the percentage of filters that are discarded when utilizing the RReLU activation function. Additionally, a comparison is made between the number of network parameters and FLOPs when using RReLU versus the conventional ReLU activation function. Finally, the paper reports the accuracy achieved after discarding unnecessary weights/filters.

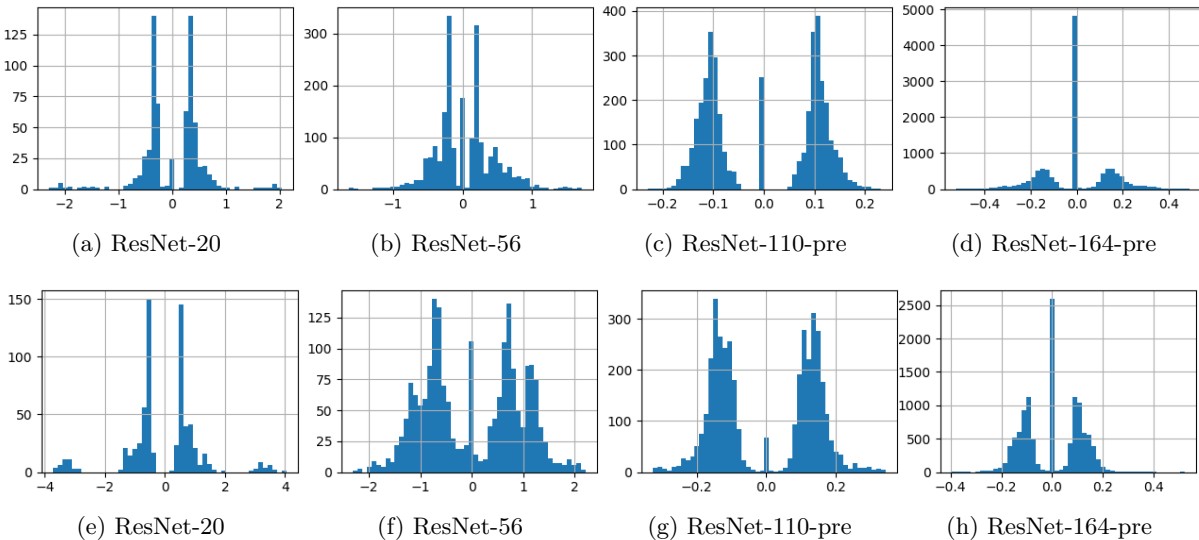

Figure 7: Distribution of RReLU slopes ($\mathbf{b}_l$) with CIFAR-10 (top) and CIFAR-100 (bottom).

The first experiment uses a fully connected neural network (FCNN) with a hidden layer of 500 neurons for MNIST classification. Both ReLU and RReLU architectures are trained for 300 epochs using the same Adam optimizer and achieve a similar range of accuracy. After training, RReLU slopes $\mathbf{b}_l$ less than a predefined threshold $\zeta$ can be made zero without degrading performance. By setting $\zeta$ by cross-validation[4], 24 out of 500 $\mathbf{b}_l$ with values less than $\zeta = 1.0$ are dropped, making 4.8% of the weight matrix at the hidden layer unnecessary. The savings are even greater when the architectures are deeper or have more parameters due to convolution operation.

With the CIFAR-10 dataset, the ResNet-164-pre architecture achieves an accuracy of 95.10% with RReLU, while ReLU achieves 95.51%. The proposed RReLU method leads one to discard 45.34% of the total filters, leading to significantly improved sparsity and computational efficiency. The number of parameters reduces from 1.7 Mn to 0.85 Mn, and the number of FLOPs reduces from 489 Mn to 309 Mn. Therefore, RReLU saves 49.1% in memory and 36.81% in computation without a regularizer. Interestingly, for a not-so-deep but wider architecture like WRN-40-4 (depth 40, widening factor 4), which has 8.9 Mn trainable parameters, the proposed RReLU activation saves 63.37% in memory and 52.20% in computation for CIFAR-10. While saving memory and computation, the WRN-40-4 architecture with RReLU does not compromise on accuracy; for example, it achieves an accuracy of 96.01% comparable to the 96.18% achieved with ReLU. The performance of the proposed RReLU activation function is evaluated on CIFAR-100 and SVHN datasets, and the outcomes are presented in Table 1, demonstrating consistently noteworthy results.

To get an idea about speedup concerning the saving in memory and FLOP reported in the paper, we have done an experiment. First, we have trained the model with RReLU which makes some of the filters insignificant after training. As RReLU can ignore a set of filters and sub-filters in two consecutive layers, for inference we can create a smaller architecture, and overwrite its parameters by the values of the significant filters from the original trained model. The smaller model performs the same as the original model but has a lesser number of filters. Therefore, the inference time of this smaller model will be less. As the inference time is a metric for showing speedup (Bianco et al., 2018), therefore we measured the total inference time of both the original model and the smaller model in a GPU device. The experiment is is detailed in Sec. A.7 of the Appendix. The corresponding inference time calculations are given in Table 10 and Table 11 in the Appendix for VGG and ResNet-164-pre, respectively. Note that the average inference time of the pruned model using the RReLU slope is lower for both VGG and ResNet-164-pre networks. We have taken a higher batch size to keep the GPU utilization in the same range across all three architectures (i) ReLU (Baseline),

---

[4]The value of $\zeta$ is picked such that the performance does not degrade after the pruning. The values we found for $\zeta$ via cross-validation are given in Table. 9 in the Appendix.

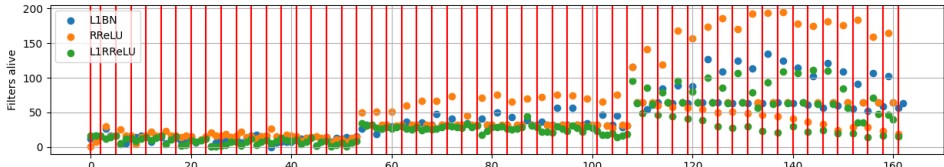

Figure 8: Filters alive at different depth for ResNet-164 and ResNet-164-RReLU

(ii) RReLU (Unpruned), and (iii) RReLU (Pruned). Keeping the utilization at the same level, we compare the inference time, which shows that RReLU (Pruned) has the minimum inference time therefore it gives the maximum speedup.

### 5.5 RReLU enhances the representation capability of the filters

To demonstrate that RReLU provides extra representation capability to batchnorm parameters, we first establish a baseline using the work by Liu et al. (2017), where the authors learn efficient convolutional networks through network slimming. We denote it by ResNet-164-L1BN, where the authors claim that $L_1$ regularization imposed on the batchnorm parameters, $\gamma_l$, leads to an increased number of $\gamma_l$ values near zero. Note that Liu et al. (2017) trained the ResNet164-L1BN model for 160 epochs, after which only 31% of the filters could be removed without any degradation in accuracy (94.8%), which resulted in 19.3% saving in memory and 26.6% saving in FLOP. We have observed earlier that sufficient training gives rise to better convergence of network parameters, resulting in better sparsity and validation accuracy. Therefore, we train the same model for 400 epochs and call it ResNet-164-L1BN-MT (MT: more training), after which 44% of the filters could be removed without any degradation in accuracy (95.1%) as reported in Table 2 (third column). In the fourth column of Table 2, without any use of a regularizer, the proposed RReLU (ResNet-164-RReLU) can prune 31.96% of the filters with even better accuracy (95.54%) than ResNet-164-L1BN-MT. While ResNet-164-L1BN-MT achieves 32.33% saving in memory and 26.38% saving in computation, the proposed ResNet-164-RReLU achieves a saving of 35.92% in memory and 25.97% in FLOP. All the models in Table 2 are trained for 400 epochs for a fair comparison. From these results, it shows that With RReLU, the models can achieves similar sparsity as L1BN-MT without using any regularizer. One may inquire as to why L1BN-MT exhibits a higher number of pruned filters but a similar degree of savings compared to RReLU. The explanation lies in the fact that toward deeper layers, RReLU primarily targets larger filters of size $3 \times 3$, whereas L1BN-MT prunes both $3 \times 3$ and $1 \times 1$ filters. In Fig. 8, the vertical red lines denote the depths where the convolution operation uses $3 \times 3$ filters while the rest of the convolution operations use filters with kernel size 1. It is evident that within the higher depths, RReLU achieves a higher degree of sparsity, resulting in greater savings in memory and FLOP compared to the number of pruned filters.

Whereas ResNet-164-RReLU has RReLU without any regularizer, ResNet-164-L1RReLU uses a $L_1$ regularizer with a sparsity regularizer of $s = 5e^{-5}$. The proposed ResNet-164-L1RReLU achieves even higher accuracy with a great saving of 64.67% in memory and 52.96% in FLOP. The proposed ResNet-164-L1RReLU achieves an accuracy of 95.25% which is very close to a ResNet-164 model trained with ReLU. As shown in Fig. 8, the ResNet-164-L1RReLU achieves higher degree of sparsity within the ranger of higher depth both in terms of bigger ($3 \times 3$) and smaller ($1 \times 1$) filters, resulting in higher savings.

| Methods | Baseline | Pruning methods | | |
|---|---|---|---|---|
| Architecture | ResNet-164 | ResNet-164-L1BN-MT (Liu et al., 2017) | ResNet-164-RReLU (Proposed) | ResNet-164-L1RReLU (Proposed) |
| Acc (with CIFAR10) | 95.77 | 95.10 | **95.54** | **95.25** |
| Filters pruned (%) | – | 44 | 31.96 | 57.44 |
| Params in Mn(% saving) | 1.67 | 1.13(32.33%) | **1.07**(35.92%) | **0.59**(64.67%) |
| FLOPs in Mn(% saving) | 489 | 360(26.38%) | **362**(25.97%) | **230**(52.96%) |

Table 2: Pruning capability of RReLU. All sparse models are trained for 400 epochs before sparsification.
Percentage values inside parentheses indicate corresponding savings.

Comparing Fig. 9b with Fig. 9a shows that without regularization, the number of elements of $\boldsymbol{\gamma}_l$ close to zero is 1600, whereas with regularization, this number has increased to 4000. Without any use of regularizer, the proposed RReLU (ResNet-164-RReLU) has 2500 elements of $\boldsymbol{\gamma}_l$ close to zero lesser than the ResNet-164-L1BN-MT, as depicted in Fig. 9c. At the end of 400 epochs, ResNet-164-L1BN-MT has more $\boldsymbol{\gamma}_l$ parameters close to zero than ResNet-164-RReLU because of the $L_1$ regularization applied to the $\boldsymbol{\gamma}_l$. The introduction of $L_1$ regularization to the RReLU slopes, denoted as ResNet-164-L1RReLU, leads to a pronounced enhancement in sparsity, as evidenced in Figure 9d. The implementation of regularization on $\mathbf{b}_l$ yields a substantial increment in the number of elements within $\boldsymbol{\gamma}_l$ that approach zero, with this count reaching 6200.

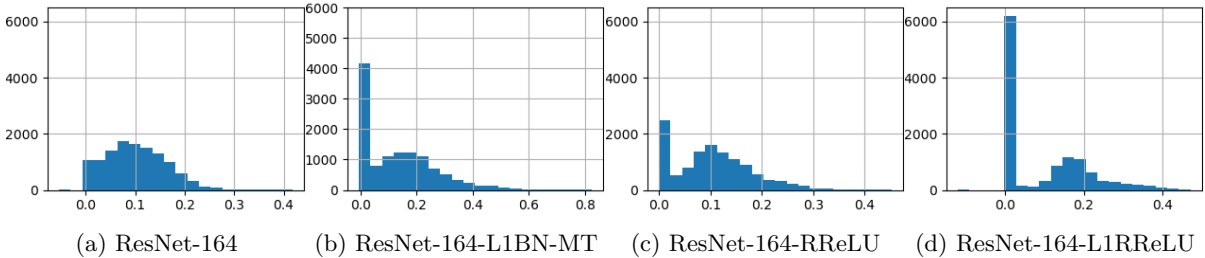

(a) ResNet-164  (b) ResNet-164-L1BN-MT  (c) ResNet-164-RReLU  (d) ResNet-164-L1RReLU

Figure 9: Effect of RReLU on BN scaling parameters $\boldsymbol{\gamma}_l$. (a) ResNet164 (b) ResNet164-L1BN-MT (c) ResNet164 with RReLU as activation (d) ResNet164 with RReLU as activation and $L_1$ regularization imposed on $\mathbf{b}_l$.

In Fig. 10, we show the type of values $\mathbf{b}_l$ and $\boldsymbol{\gamma}_l$ take after training ResNet-164-L1RReLU. Out of 54 residual blocks, we pick 4 blocks at random and plot $\mathbf{b}_l$ (y-axis) against $\boldsymbol{\gamma}_l$ (x-axis) [5]. Given that the regularization is applied to $\mathbf{b}_l$, it compels these parameters to adopt smaller values. However, as the elements of $\boldsymbol{\gamma}_l$ are not regularized and can hence assume a wide range of values, the term $\boldsymbol{\gamma}_l\mathbf{b}_l$ can take any value in the real line as shown in the figures. In every block, we see that when $\mathbf{b}_l$ takes significant values, $\boldsymbol{\gamma}_l$ takes a range of values, giving a higher representation power to every filter. Therefore, in our approach, we prune the filters only based on the values of $\mathbf{b}_l$. If we look at Fig. 9b for ResNet-164-L1BN-MT, we see that, when $L_1$ regularization is applied on $\boldsymbol{\gamma}_l$, many of the batchnorm parameters $\boldsymbol{\gamma}_l$ are pushed to take values in the range $0 < \gamma_l^i \leq 0.1$. However, looking at the same range in Fig. 9c and 9d while using RReLU, we see that the number of elements of $\boldsymbol{\gamma}_l$ in that range is less. These values tend to either shift towards higher magnitudes or concentrate around values close to zero, indicating more flexibility with RReLU.

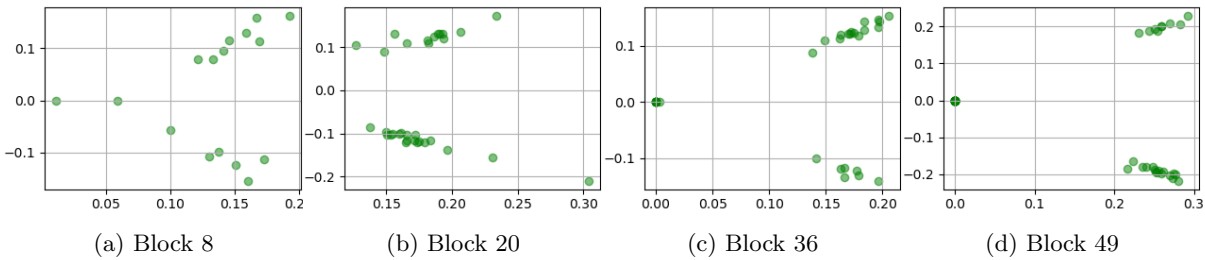

(a) Block 8  (b) Block 20  (c) Block 36  (d) Block 49

Figure 10: Plot of RReLU slopes ($\mathbf{b}_l$) along y-axis vs. BN parameters ($\boldsymbol{\gamma}_l$) along x-axis in different residual blocks of ResNet-164-L1RReLU.

Also note that, with more depth, more number of filters could be pruned as more number of ($\mathbf{b}_l, \boldsymbol{\gamma}_l$) is close to zero [6]. For example, in $8^{th}$ residual block (Fig. 10a), which consists of shallow layers, there are relatively few slopes in proximity to zero, limiting the pruning capacity. In contrast, when examining deeper layers,

---

[5]In Fig. 17 in the Appendix, we show such plots for more blocks.
[6]This trend is visible in Fig. 17 of the Appendix.

such as the 49th block, numerous pairs of $(\mathbf{b}_l, \boldsymbol{\gamma}_l)$ gravitate towards zero, indicating a greater potential for filter pruning.

Following equation 3, one can observe that when the $L_1$ regularization is applied to the batchnorm parameters $\boldsymbol{\gamma}_l$, it enforces all the elements of $\boldsymbol{\gamma}_l$ to take lower values. This constraint may limit the representation capacity of the ResNet-164-L1BN-MT model. But with RReLU, there is no such restriction on $\boldsymbol{\gamma}_l$ or $\mathbf{b}_l$, and RReLU provides another degree of freedom by the slopes $\mathbf{b}_l$. Therefore, When $\mathbf{b}_l$ is significant, the elements of $\boldsymbol{\gamma}_l$ take an unconstrained range of values, e.g. in Fig. 10b. It improves the representation power of the corresponding filters. Therefore, it inherently enables a greater number of parameters of $\mathbf{b}_l$ to approach zero, leading to disregarding corresponding filters, all achieved without the need for explicit regularization, thus resulting in implicit sparsification.

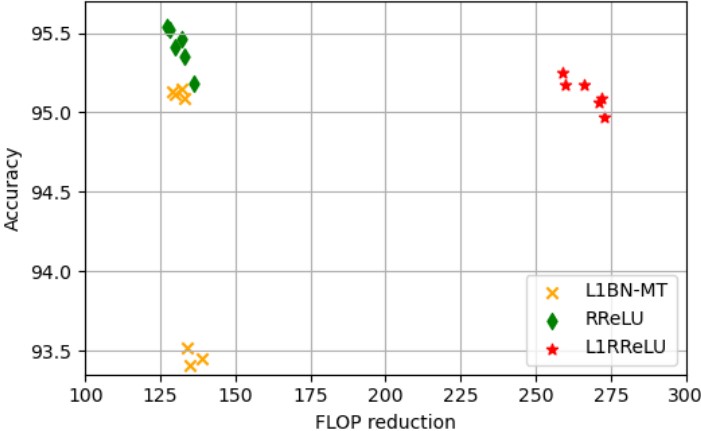

Figure 11: Acc vs FLOP reduction. The proposed methods (RReLU, L1RReLU) are compared with L1BN-MT (MT: more training)

In Fig. 11, we present the top 1% accuracy vs. FLOP reduction trade-off for the ResNet-164 model. The FLOP reduction signifies the difference between the FLOP count of the ResNet-164 and the FLOP count of the models employing various sparsifying methods, as depicted in Figure 11. Enhanced performance is indicated by a positioning closer to the upper-right quadrant of the graph, as it indicates higher accuracy while using lower no of FLOPs. Without using a regularizer, the RReLU provides same FLOP reduction as L1BN-MT with a little higher accuracy, indicating that RReLU provides intrinsic sparsification. Further, when $L_1$ regularization is applied on RReLU slopes, L1RReLU provides the best FLOP reduction with a same level of accuracy.

With its excellent sparsification capability, decomposed training (Yang et al., 2020a) achieves exceptional speedup, keeping the accuracy the same. Whereas decomposed training maintains an accuracy of 93.28% for ResNet56, we achieve an accuracy of 94.11% with RReLU (without any regularization). As RReLU demonstrates inherent sparsification properties, it can be seamlessly integrated into the architectures employed in the study by Yang et al. (2020a) in conjunction with decomposed training techniques. This integration has the potential to yield significantly improved speedup and enhanced accuracy. In the below paragraph, we show it can be applied to another recent structural pruning work by Hou et al. (2022).

### 5.6   RReLU combined with other structural pruning methods improves sparsity

Here we show that when RReLU is combined with the structural pruning methods such as CHEX Hou et al. (2022), the sparsity further improves. The pruning and regrowing for the CHEX method are performed during the training after every $\Delta T = 2$ time step. CHEX considers ResNet50 for performing classification on the ImageNet Dataset. ResNet50 on the ImageNet classification task achieves an accuracy of 76.13% using a GFLOP count of 4.09. The ResNet50 architecture with RReLU reduces the GFLOP to 3.45 without using any regularizer. This shows that RReLU can provide sparsity intrinsically. Here, both the architectures with

ReLU and RReLU, respectively, are trained for 250 epochs. We have observed that instead of the usual multi-step learning rate (LR) scheduler, if the cosine annealing LR scheduler is used, then the accuracy improves to 76.28% with 3.15 GFLOPs. The improved performance indicates that if trained well with appropriate hyperparameters and proper initialization, the architectures with RReLU can have better accuracy than ReLU with a lesser number of GLOPs within the same number of time steps.

We acknowledge that compared to recent sparsity-inducing methods such as CHEX, RReLU provides less sparsity. However, it's crucial to recognize that the sparsity induced by RReLU is intrinsic. As highlighted in our paper, RReLU is an activation function that inherently promotes sparsity and is not a pruning technique. Consequently, RReLU can be effectively combined with various structured pruning methods to enhance sparsity, as each filter has better representation power with RReLU. To validate this, we have integrated RReLU with CHEX and compared the GFLOP counts for ResNet50 with ReLU and RReLU below.

| Methods | Accuracy | GFLOPs |
|---|---|---|
| ReLU (multistep LR) | 76.13% | 4.09 |
| RReLU (multistep LR) | 75.22% | 3.45 |
| RReLU (cosine annealing) | **76.28**% | 3.15 |
| CHEX w/ ReLU ($s = 70\%$, cosine annealing) | 76.28% | 1.35 |
| CHEX w/ RReLU ($s = 70\%$, cosine annealing) | 75.48% | **1.12** |
| CHEX w/ ReLU ($s = 50\%$, cosine annealing) | 77.98% | 2.49 |
| CHEX w/ RReLU ($s = 50\%$, cosine annealing) | 77.58% | **2.20** |

Table 3: Comparison between (i) ReLU (ii) RReLU (ii) CHEX, and (iv) RReLU combined with structured pruning method CHEX for ResNet50 on ImageNet dataset.

With a target sparsity of $s = 70\%$, the baseline (CHEX w/ ReLU) with ResNet50 on ImageNet classification achieved an accuracy of 76.28% with a GFLOP count of 1.35 when trained from scratch using the codebase and the hyperparameters the authors have provided[7]. When CHEX is combined with RReLU (CHEX w/ RRELU), the architecture reduces the computation to 1.12 GFLOPs, as shown in Table. 3, corroborating our claim. Further, with a target sparsity of $s = 50\%$, CHEX reduces the computation from 4.09 GFLOPs to 2.20 GFLOPs, but when CHEX is combined with RReLU, it reduces the computation to 2.20 GFLOPS. This shows that RReLU improves the sparsity when combined with existing structured pruning techniques. The detailed results are provided in Table. 3.

### 5.7 RReLU as the coarse feature extractor

During training, the learned weights/filters are tuned to extract finer features of the input. The RReLU activation function is observed to extract the coarse features where the weights/filters are random. In order to substantiate this claim, we conducted an experiment whereby only the RReLU slopes ($\mathbf{b}_l$) are trained for 500 epochs. At the same time, the weights are fixed, following initialization with the Kaiming He method (He et al., 2015). The outcomes indicate that while a neural network architecture with untrained weights and a ReLU activation function fails to learn anything, an architecture with solely learned $\mathbf{b}_l$ can attain some degree of learning. Specifically, ResNet-56 with only the learned $\mathbf{b}_l$ (and untrained weights) yielded an accuracy of 51.42% with the CIFAR-10 dataset, and one can remove 7.09% of the filters using $\zeta = 0.1$, as shown in Table 4.

These findings suggest that the RReLU activation function, particularly its learned slopes, extracts the coarse features of the inputs in terms of the untrained filters in the case of CNNs. Fig. 12 shows the histogram of the RReLU slopes of ResNet-56 after training. Once the RReLU is used as a coarse feature extractor, one can drop those RReLU that have nearly zero slopes and prune their corresponding filters. We can then train the remaining weights and retrain the slopes to attain the finer features. After the $2^{nd}$ step, the models tend to achieve similar accuracy as the baseline, as seen in Table 4.

---

[7]https://github.com/zejiangh/Filter-GaP/tree/main

| Dataset | CIFAR-10 | | CIFAR-100 | |
|---|---|---|---|---|
| Architecture | ResNet20 | ResNet56 | ResNet20 | ResNet56 |
| Acc ReLU (standard) | 93.03 | 93.63 | 68.20 | 69.99 |
| Acc ($1^{st}$-step) | 45.12 | 51.42 | 8.02 | 10.54 |
| Filters ignored (%) | 5.35 | 7.09 | 6.84 | 7.44 |
| #Params RReLU | 0.24 | 0.72 | 0.23 | 0.72 |
| #FLOPs RReLU | 72.5 | 220 | 70.29 | 216 |
| Acc ($2^{nd}$ step) | 91.93 | 92.82 | 66.68 | 66.39 |

Table 4: RReLU extracts the coarse features with $\mathbf{b}_l$ being only the trainable parameters.

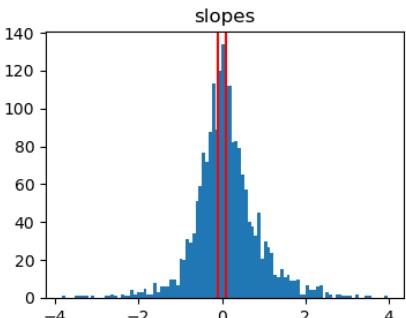

Figure 12: Distribution of $\mathbf{b}_l$ for ResNet-56 with CIFAR-10.

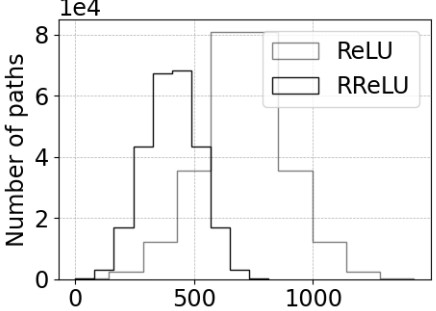

Figure 13: Distribution of filter-path length for WRN-40-4 with CIFAR-100.

## 5.8 Features choose the shortest path length

Veit et al. (2016) demonstrate residual networks as an ensemble of independent paths, and the study show that only shorter paths carry gradients despite using deeper architecture. Our study supports these findings as we observe that features not only try to pass through shorter paths, but they try to pass through a lesser number of filters as well. While Veit et al. (2016) define path-length as the number of layers the input features pass through, we define *filter-path length* as the number of filters the features pass through. For a WRN-40-4 with ReLU activation, most paths have a filter-path length between 300 to 1100, but using RReLU, we observe a reduction in filter-path length, with most paths having lengths between 100 to 500, as shown in Fig. 13. This shows that RReLU activation allows features to pass through fewer filters, resulting in saved memory and computation.

## 5.9 Scalable across larger dataset and various architectures

In this part, we show results with the benchmark ILSVRC-2012 (ImageNet-1k) dataset in Table. 5. For testing with ImageNet-1k, we consider three architectures: (i) a WideResNet with depth 50 and widening factor 2 denoted by WRN-50-2, (ii) a Vision Transformer with 12 layers, a width of 384, the MLP width of 1536, and patch-size of 16 denoted by VIT-s16 (Steiner et al., 2021), and an (iii) EfficientNet. For training WRN-50-2, we closely follow the official implementation[8] for image classification with ImageNet dataset with standard PyTorch models. We have selected WRN-50-2 due to its superior performance compared to considerably deeper ResNets such as ResNet-152. We do not use pre-trained parameters for initializing the network and train WRN-50-2 for both ReLU and RReLU from scratch. In both scenarios, whether utilizing ReLU or RReLU activation functions, the architectures of WRN-50-2 undergo training for a total of 120 epochs, the same as the one considered in the official PyTorch repository.

VIT-s16 was successfully trained on ImageNet by Dosovitskiy et al. (2020), attaining excellent results compared to familiar convolutional architectures. The VIT-s16 architecture uses only the Encoder block of a

---

[8] https://github.com/pytorch/examples/tree/main/imagenet

Transformer, which consists of 12 blocks, each having a Multi-head-dot-product-attention, a multi-layer perceptron (MLP) block and two Dropout layers and two Layer-normalization layers [9] Even though Dosovitskiy et al. (2020) proposed Vision Transformer, they published codes only for fine-tuning and pre-trained models. In a subsequent study by Steiner et al. (2021), the researchers conducted a thorough investigation into the relationship between the amount of training data, model regularization/data augmentation, model size, and compute budget. They made their code base available with a configuration (Beyer et al., 2022) to train plain VIT baselines from scratch for ImageNet-1k. In our research, we utilized this framework to present results with the VIT-s16 model. This plain VIT baseline uses standard data augmentation and no sophisticated regularization techniques such as augmentation or regularization pipeline. It does not use any pre-training procedure. The VIT-s16 model for ReLU is trained for 300 epochs, whereas for better convergence of RReLU slope, ViT-s16 with RReLU is trained for 400 epochs.

EfficientNet has shown superior performance post-ResNet and is famous for efficient computation. For example, for a similar accuracy, EfficientNet-b1 is $7.6x$ smaller and $5.7x$ faster than ResNet-152 Tan & Le (2019). EfficientNet uses (i) depthwise separable convolution and (ii) sequential squeeze and excitation in an inverted bottleneck residual block along with intermediate expansion operation and therefore is a better version than MobileNet which has only depthwise separable convolution. Therefore, we proceeded with comparisons only against EfficientNet, as it represents a superior architecture than MobileNet Howard et al. (2017) for studying the performance of RReLU. EfficientNet uses Sigmoid Linear Unit (SiLU) activation. We show the intrinsic sparsity achieved by using the idea of rotation in one of the EfficientNet architecture i.e. EfficientNet-b4. We replace SiLU with rotated SiLU (RSiLU) and train it. The EfficientNet architectures with ReLU are trained for 400 epochs, whereas we report the performance of EfficientNet with RSiLU after 500 epochs.

| Arc | Activation | Filters ignored (%) | Accuracy | Params(Mn) | Pruned | FLOP(Mn) | Pruned |
|---|---|---|---|---|---|---|---|
| WRN-50-2 | ReLU | - | 76.682 | 67.4 | - | 21563 | - |
| WRN-50-2 | RReLU | **25.34** | 76.58 | **55.2** | 18.1% | **18648** | 13.5% |
| VIT-s16-MLP | GELU | - | 77.51 | 14.2 | - | 28.3 | - |
| VIT-s16-MLP | RGELU | 6.32 | **80.1** | **13.2** | 6.32% | **26.5** | 6.32% |
| EfficientNet-b4 | SiLU | - | 82.80 | 19.3 | - | 4200 | - |
| EfficientNet-b4 | RSiLU | 16.83% | 81.67 | **17.2** | 10.9% | **3500** | 16.66% |

Table 5: Applying Rotation on ReLU, GELU, and SiLU activation with Imagenet dataset.

While WRN-50-2 architecture uses ReLU, each MLP block of VIT-s16 contains two dense layers with a GELU non-linearity in between two layers. So, for WRN-50-2, we have used RReLU instead of ReLU; for VIT-s16, we have used RGELU instead of GELU. In Table 5, we demonstrate the sparsity property of RReLU or RGELU. Training WRN-50-2 with RReLU for ImageNet demonstrated the implicit regularization property of RReLU, allowing for the removal of 25.34% of the filters. This led to a 18.1% reduction in parameters and a 13.5% reduction in FLOPs, keeping the performance the same. The VIT-s16 architecture boasts a total parameter count of 22.14 million, with 14.2 million parameters dedicated to the MLP component, while the remaining parameters are attributed to the multihead-dot-product attention mechanism. So in Table 5, we report the saving in the number of parameters and FLOPs in VIT-s16-MLP. Whereas the VIT-s16 was reported to achieve an accuracy of 77.51% with GELU, the same with RGELU achieved an accuracy of 80.1% with sufficient training. Not only does the accuracy improve, but RGELU provides a saving of 6.32% in both memory and FLOPs. The baseline architecture, EfficientNet-b4 has 19.3 Mn parameters and 4.2 Bn FLOPs and is reported to have an accuracy of 82.80%. Using RSiLU provides an intrinsic sparsity to reduce the parameter count and FLOP count to 17.2 Mn and 3.5 Bn, respectively, with a comparable accuracy of 81.67%. We could run only one iteration because of limited computing power, but we believe that the accuracy can be improved with better hyperparameter tuning. These results imply that rotation operation is transferable to other activations if the activation mimics ReLU.

---

[9]For detailed architecture, please refer to `https://github.com/google-research/big_vision`.

### 5.10 RReLU imposes better adversarial robustness

From the distribution of $\mathbf{b}_l$ in Fig. 7, it is observed that most of the RReLU slopes have values less than one in most cases. As LC can be defined as the upper bound on the gradient of the function $f(x)$ (Latorre et al., 2020; Paulavičius & Žilinskas, 2006), it can be argued that the value of LC is typically lower for RReLU than ReLU, resulting in better adversarial robustness. However, in practice, calculating the Lipschitz Constant (LC) for the neural network is NP-hard, and an alternative empirical computation of the Local Lipschitz Constant (LLC) is preferred and defined as the following quantity (Yang et al., 2020b)

$$\frac{1}{n} \sum_{i=1}^{n} \max_{\mathbf{x}'_i \in \mathbb{B}_\infty(\mathbf{x}_i, \epsilon)} \frac{||f(\mathbf{x}_i) - f(\mathbf{x}'_i)||_{KL}}{||\mathbf{x}_i - \mathbf{x}'_i||_\infty}. \tag{12}$$

To perform the max operation in equation 12, we adopt a gradient-descent-like algorithm with a step-

| Architecture | activation | LLC | Attack type | Adv test acc |
|---|---|---|---|---|
| ResNet-20 | ReLU | 1.33 | FGSM
PGD | 34.11
38.20 |
| | RReLU | 1.2 | FGSM
PGD | **39.84**
**42.46** |
| ResNet-56 | ReLU | 1.41 | FGSM
PGD | 59.01
16.45 |
| | RReLU | 1.30 | FGSM
PGD | **66.85**
**17.05** |

Table 6: RReLU to boost local smoothness and hence adversarial accuracy.

size of 0.001 and 60 iterations. We have computed the LLC using equation 12 and provided in the third column of Table 6, which shows that the LLC for the architectures with RReLU is lower than the LLC for the architectures with ReLU, indicating better smoothness (Yang et al., 2020b). We consider a single-step attack, such as the Fast Gradient Sign Method (FGSM), and an iterative attack, Projected Gradient Descent (PGD), to test the adversarial robustness of architectures with RReLU. FGSM samples considered in our experiments have a perturbation of $\epsilon = 0.031$. The input variation parameters $\epsilon_v$ for the PGD attack are considered to be 0.031 and 0.1 for ResNet20 and ResNet56, respectively. The step size for each attack iteration is $\epsilon_v/20$, and the number of attack iterations is 10. Observing the adversarial test accuracy (for both FGSM and PGD) at the last column of Table 6, we see that for the type of attack we have considered, the architectures with RReLU have better robustness than one with ReLU.

### 5.11 A study on the sparsification capability of batchnorm scaling parameters versus RReLU slopes

Mathematically, it looks like the RReLU slopes can be learned by the outer convolution operation. But in the absence of RReLU slopes, just the batchnorm parameters alone cannot render some filters unnecessary to achieve structural sparsity. We suspect the reason is that the batchnorm parameters are usually initialized with a positive value; therefore, they do not explore negative values well. Batchnorm parameters cannot be initialized with a range of both positive and negative values because with negative values, during training, there is a high chance that the output from the batchnorm gets mapped to the negative part of the ReLU slope and the information does not pass to the next layer.

To understand the relation between the batchnorm parameters $\boldsymbol{\gamma}_l$ and RReLU slopes $\mathbf{b}_l$, we have done the following experiments. We study the effect of both $\boldsymbol{\gamma}_l$ and $\mathbf{b}_l$ in performance in terms of both accuracy and sparsity. So, we consider four cases:

- The value of the elements of $\mathbf{b}_l$ is fixed to 1 which is nothing but ReLU and we train only $\boldsymbol{\gamma}_l$. This is the baseline, i.e., experiment with ReLU.

- The value of every element of $\boldsymbol{\gamma}_l$ is fixed to the initial value of 0.5 and train only $\mathbf{b}_l$.

- We fix elements of both $\boldsymbol{\gamma}_l$ and $\mathbf{b}_l$ to 0.5 and 1, respectively.

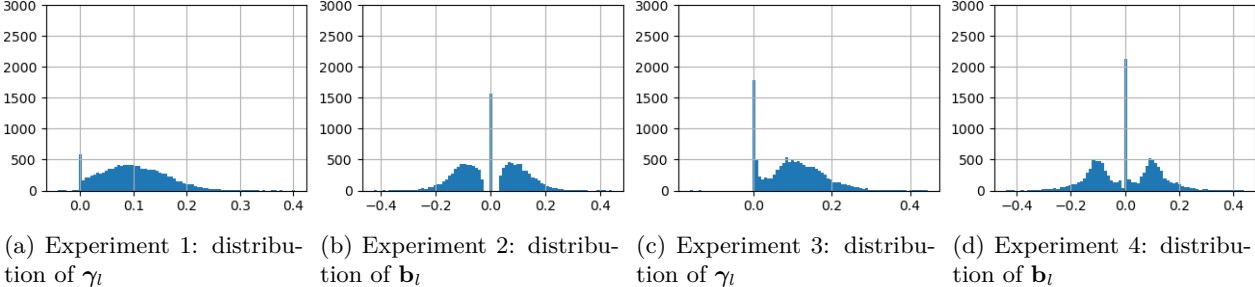

(a) Experiment 1: distribution of $\boldsymbol{\gamma}_l$

(b) Experiment 2: distribution of $\mathbf{b}_l$

(c) Experiment 3: distribution of $\boldsymbol{\gamma}_l$

(d) Experiment 4: distribution of $\mathbf{b}_l$

Figure 14: Distributions of $\boldsymbol{\gamma}_l$ and $\mathbf{b}_l$

- We train both $\boldsymbol{\gamma}_l$ and $\mathbf{b}_l$.

All the above models are trained for 400 epochs without using any regularizer. Below are our observations:

- The first case is nothing but the baseline where we use a model with ReLU and the $\boldsymbol{\gamma}_l$ is trainable. We achieved an accuracy of 95.79. In this case, only 755 elements of $\boldsymbol{\gamma}_l$ are close to zero as we do not use $L_1$ regularization on $\boldsymbol{\gamma}_l$. The distribution of elements of $\boldsymbol{\gamma}_l$ is shown in Fig. 14a. So it is clear that the trainable $\boldsymbol{\gamma}_l$ is not able to induce sparsity much by itself.

- In the second case, we fix every element of $\boldsymbol{\gamma}_l$ to 0.5 and learn the RReLU slopes $\mathbf{b}_l$ along with other trainable network parameters. The aim is to understand if $\mathbf{b}_l$ can do the job of $\boldsymbol{\gamma}_l$. We achieve an accuracy of 95.70, close to the previous baseline. From this, we can see that even if we keep the $\boldsymbol{\gamma}_l$ fixed, the $\mathbf{b}_l$ are still able to do the job of $\boldsymbol{\gamma}_l$. So, it is not necessary to have $\boldsymbol{\gamma}_l$ to reach a similar accuracy as the first case. The distribution of the elements of $\mathbf{b}_l$ are shown in Fig. 14b. One can observe that the number of slopes close to zero is 1561, which is higher than the previous case where the number of elements of $\boldsymbol{\gamma}_l$, close to zero, is 755.

- In the third case, we fix both $\boldsymbol{\gamma}_l$ and $\mathbf{b}_l$ to see if the network filter weights and biases are enough to reach the same accuracy as the above two cases. We observe that it reaches an accuracy of 95.40, almost the same as the previous two methods. But please note that as the elements of $\boldsymbol{\gamma}_l$ and $\mathbf{b}_l$ are fixed to values 0.5 and 1.0, respectively, the sparsity could not be induced.

- The fourth case has both $\boldsymbol{\gamma}_l$ and $\mathbf{b}_l$ are trainable and the model reaches an accuracy of 95.84. The distributions of $\boldsymbol{\gamma}_l$ and $\mathbf{b}_l$ are shown in Fig. 14c and Fig. 14d, respectively. When both $\boldsymbol{\gamma}_l$ and $\mathbf{b}_l$ are trained together, the sparsity is more. We see that 2354 slopes are near zero. But a slightly lesser number of $\boldsymbol{\gamma}_l$ (2282) are close to zero because when the elements of $\mathbf{b}_l$ do not take values close to zero, the elements of $\boldsymbol{\gamma}_l$ take a wide range of values. This is more prominent when the network is trained for more epochs. Comparing the Fig. 14d with Fig. 14b, one can observe that with trainable $\boldsymbol{\gamma}_l$ and $\mathbf{b}_l$, the sparsity is more than only trainable $\mathbf{b}_l$ even though $\mathbf{b}_l$ can do the job of $\boldsymbol{\gamma}_l$ for achieving good accuracy. Comparing Fig. 14c with Fig. 14a shows that with trainable $\boldsymbol{\gamma}_l$ and $\mathbf{b}_l$, the number of $\boldsymbol{\gamma}_l$ close to zero also increases. Further, the sparsity increases if the model is trained for more training epochs as more elements of $\mathbf{b}_l$ go towards zero. We have seen that once trained for 1200 epochs, the sparsity is even more.

We have also shown how the distributions change during optimization for some of the $\boldsymbol{\gamma}_l$ in Fig. 18 of the Appendix. The elements of $\boldsymbol{\gamma}_l$ are initialized with 0.5, and during training, some of the elements of $\boldsymbol{\gamma}_l$ those which correspond to the zeroed elements of $\mathbf{b}_l$ (during training), slowly converge to take values close to zero. Fig. 19a, Fig. 19b, and Fig. 19c in the Appendix are showing the distribution of $\mathbf{b}_l$ in the first, second, and third layer, respectively of every residual block. Over time, some of the elements of $\mathbf{b}_l$ move towards zero.

# 6 Conclusion

In this work, we proposed the RReLU activation and a novel initialization for the RReLU slopes. We then demonstrated through extensive experiments that the combination plays a vital role in the implicit sparsification of the architecture by enhancing the representation capability of the architecture. The proposed activation saves resources and improves performance in many architectures. For example, with the CIFAR-10 dataset, the WRN-40-4 achieves 63.37% saving in memory and 69.7% saving in computation without any loss in accuracy. The proposed RReLU can be used as an alternative to ReLU in all deep architectures where ReLU had been used earlier including other structural pruning methods too. The RReLU-based architectures are also shown to have a lower LLC value and hence better adversarial robustness. Therefore, RReLU is a one-stop solution for improved performance, energy efficiency, and robustness. The idea of learning the rotation of the linear component of the activation function can also be applied to other activation functions which have a partly linear component, i.e. we have applied the proposed idea to GELU and observed that RGELU in ViT-s16 achieves an accuracy better than GELU with 6.32% saving in memory and computation of the MLP block.

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

## A  Appendix

In this section we provide experiment details so that it is easier for the readers to reproduce the results.

### A.1  The sample ResNet and WideResNet architectures considered for our experiments

The architecture of vanilla ResNet-20 is given in Table. 7 where each of the three blocks has 6 layers. For vanilla ResNet-56, each block has 9 layers He et al. (2016a); Idelbayev. To obtain the architecture with RReLU, the ReLU activations at all the layers (except the first convolution layer) are converted to RReLU. We do not replace the ReLU after the first convolution layer with RReLU because we may not want to miss out on any information in the input by deactivating filters at the very first layer. For CIFAR-10 dataset, $K = 10$ and for CIFAR-100, $K = 100$.

| Layer | Description | Output Dimension |
|---|---|---|
| Input | - | $3 \times 32 \times 32$ |
| Conv | $C_{in} = 3, C_{out} = 16$, Kernel: $(3 \times 3)$, BN, ReLU | $16 \times 32 \times 32$ |
| Block1-layer1 | $C_{in} = 16, C_{out} = 16$, Kernel: $(3 \times 3)]$, BN, ReLU | $16 \times 32 \times 32$ |
| Block1-layer2 | $C_{in} = 16, C_{out} = 16$, Kernel: $(3 \times 3)]$, BN, ReLU, addition | $16 \times 32 \times 32$ |
| Block1-layer3 | $C_{in} = 16, C_{out} = 16$, Kernel: $(3 \times 3)]$, BN, ReLU | $16 \times 32 \times 32$ |
| Block1-layer4 | $C_{in} = 16, C_{out} = 16$, Kernel: $(3 \times 3)]$, BN, ReLU, addition | $16 \times 32 \times 32$ |
| Block1-layer5 | $C_{in} = 16, C_{out} = 16$, Kernel: $(3 \times 3)]$, BN, ReLU | $16 \times 32 \times 32$ |
| Block1-layer6 | $C_{in} = 16, C_{out} = 16$, Kernel: $(3 \times 3)]$, BN, ReLU, addition | $16 \times 32 \times 32$ |
| Block2-layer1 | $C_{in} = 16, C_{out} = 32$, Kernel: $(3 \times 3)]$, BN, ReLU | $32 \times 16 \times 16$ |
| Block2-layer2 | $C_{in} = 32, C_{out} = 32$, Kernel: $(3 \times 3)]$, BN, ReLU, addition | $32 \times 16 \times 16$ |
| Block2-layer3 | $C_{in} = 32, C_{out} = 32$, Kernel: $(3 \times 3)]$, BN, ReLU | $32 \times 16 \times 16$ |
| Block2-layer4 | $C_{in} = 32, C_{out} = 32$, Kernel: $(3 \times 3)]$, BN, ReLU, addition | $32 \times 16 \times 16$ |
| Block2-layer5 | $C_{in} = 32, C_{out} = 32$, Kernel: $(3 \times 3)]$, BN, ReLU | $32 \times 16 \times 16$ |
| Block2-layer6 | $C_{in} = 32, C_{out} = 32$, Kernel: $(3 \times 3)]$, BN, ReLU, addition | $32 \times 16 \times 16$ |
| Block3-layer1 | $C_{in} = 32, C_{out} = 64$, Kernel: $(3 \times 3)]$, BN, ReLU | $64 \times 8 \times 8$ |
| Block3-layer2 | $C_{in} = 64, C_{out} = 64$, Kernel: $(3 \times 3)]$, BN, ReLU, addition | $64 \times 8 \times 8$ |
| Block3-layer3 | $C_{in} = 64, C_{out} = 64$, Kernel: $(3 \times 3)]$, BN, ReLU | $64 \times 8 \times 8$ |
| Block3-layer4 | $C_{in} = 64, C_{out} = 64$, Kernel: $(3 \times 3)]$, BN, ReLU, addition | $64 \times 8 \times 8$ |
| Block3-layer5 | $C_{in} = 64, C_{out} = 64$, Kernel: $(3 \times 3)]$, BN, ReLU | $64 \times 8 \times 8$ |
| Block3-layer6 | $C_{in} = 64, C_{out} = 64$, Kernel: $(3 \times 3)]$, BN, ReLU, addition | $64 \times 8 \times 8$ |
| Linear | classification into K classes with softmax | $K \times 1$ |

Table 7: Architecture of vanilla ResNet-20; For the RReLU version, the ReLU activations are replaced by ReLU. Here addition implies the addition of a skip connection with the residual path. For CIFAR-10, $K = 10$ and for CIFAR-100, $K = 100$.

The ResNet-110-pre architecture has an identity connection between two ResNet units; the batchnorm layers are followed by RReLU and the RReLU layers are followed by the convolution layers. The "pre" in its name indicates preactivation, i.e., ReLU/RReLU activation before the convolution layer. The ResNet-164-pre also has the above identity connection and preactivation; along with a special bottleneck architecture which has a different number of channels along the depth than the standard ResNets He et al. (2016b); Liu.

The WideResNetKuen architecture WRN-16-4 is given in Table. 8 that is used with the SVHN dataset. For SVHN $K = 10$. The other WideResNet architecture is WRN-40-4, which has 40 layers and a widening factor of 4. It has the same three blocks but each with 12 layers. WR N-40-4

### A.2  How to set $\zeta$ during deployment of the models to discard the slopes that are below $\zeta$ and still not degrade the performance?

The value of $\zeta$ is set by cross-validation. The network is trained with the train set, and the test set is divided into two parts. The first part of the test set is used to find a $\zeta$ such that if the slopes that are below that $\zeta$ are made equal to zero, the performance does not degrade. The number of RReLU slopes that can be ignored for different experiments and the corresponding $\zeta$ is given in Table 9. With the latter part of the

| Layer | Description | Output Dimension |
|---|---|---|
| Input | - | $3 \times 32 \times 32$ |
| Conv | $C_{in} = 3, C_{out} = 16$, Kernel: $(3 \times 3)$, BN, ReLU | $16 \times 32 \times 32$ |
| Block1-layer1 | $C_{in} = 16, C_{out} = 64$, Kernel: $(3 \times 3)$], BN, ReLU | $64 \times 32 \times 32$ |
| Block1-layer2 | $C_{in} = 64, C_{out} = 64$, Kernel: $(3 \times 3)$], BN, ReLU, addition | $64 \times 32 \times 32$ |
| Block1-layer3 | $C_{in} = 64, C_{out} = 64$, Kernel: $(3 \times 3)$], BN, ReLU | $64 \times 32 \times 32$ |
| Block1-layer4 | $C_{in} = 64, C_{out} = 64$, Kernel: $(3 \times 3)$], BN, ReLU, addition | $64 \times 32 \times 32$ |
| Block2-layer1 | $C_{in} = 64, C_{out} = 128$, Kernel: $(3 \times 3)$], BN, ReLU | $128 \times 16 \times 16$ |
| Block2-layer2 | $C_{in} = 128, C_{out} = 128$, Kernel: $(3 \times 3)$], BN, ReLU, addition | $128 \times 16 \times 16$ |
| Block2-layer3 | $C_{in} = 128, C_{out} = 128$, Kernel: $(3 \times 3)$], BN, ReLU | $128 \times 16 \times 16$ |
| Block2-layer4 | $C_{in} = 128, C_{out} = 128$, Kernel: $(3 \times 3)$], BN, ReLU, addition | $128 \times 16 \times 16$ |
| Block3-layer1 | $C_{in} = 128, C_{out} = 256$, Kernel: $(3 \times 3)$], BN, ReLU | $256 \times 8 \times 8$ |
| Block3-layer2 | $C_{in} = 256, C_{out} = 256$, Kernel: $(3 \times 3)$], BN, ReLU, addition | $256 \times 8 \times 8$ |
| Block3-layer3 | $C_{in} = 256, C_{out} = 256$, Kernel: $(3 \times 3)$], BN, ReLU | $256 \times 8 \times 8$ |
| Block3-layer4 | $C_{in} = 256, C_{out} = 256$, Kernel: $(3 \times 3)$], BN, ReLU, addition | $256 \times 8 \times 8$ |
| Linear | classification into K classes with softmax | $K \times 1$ |

Table 8: Architecture of WRN-16-4. For CIFAR-10, $K = 10$ and for CIFAR-100, $K = 100$.

test set, the pruned network is tested, and accuracy is reported in Table 1 of the main manuscript. For Imagenet and WRN-50-2, the threshold $\zeta$ is set equal to 0.045 via cross validation.

Table 9: The value of $\zeta$ for different experiments and the corresponding number of filters ignored in the 'x/y' format. Here 'x' filters out of 'y' filters could be dropped because of RReLU.

| Dataset | MNIST | CIFAR-10 | | | | | CIFAR-100 | | | | | SVHN |
|---|---|---|---|---|---|---|---|---|---|---|---|---|
| Architecture | FCNN | ResNet-20 | ResNet-56 | ResNet-110-pre | ResNet-164-pre | WRN-40-4 | ResNet-20 | ResNet-56 | ResNet-110-pre | ResNet-164-pre | WRN-40-4 | WRN-16-4 |
| $\zeta$ | 1.0 | 0.1 | 0.06 | 0.045 | 0.08 | 0.04 | 0 | 0.08 | 0.04 | 0.04 | 0.04 | 0.04 |
| Filters ignored | $\frac{24}{500}$ | $\frac{26}{672}$ | $\frac{179}{2016}$ | $\frac{241}{3984}$ | $\frac{5376}{11856}$ | $\frac{2227}{5136}$ | $\frac{0}{672}$ | $\frac{208}{2016}$ | $\frac{75}{3984}$ | $\frac{2684}{11856}$ | $\frac{924}{5136}$ | $\frac{336}{1552}$ |

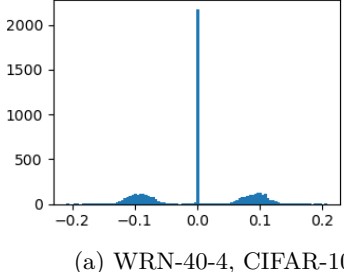

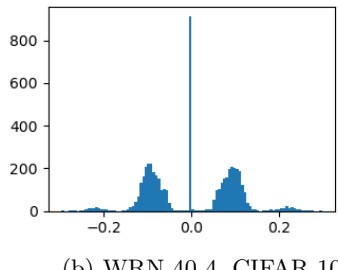

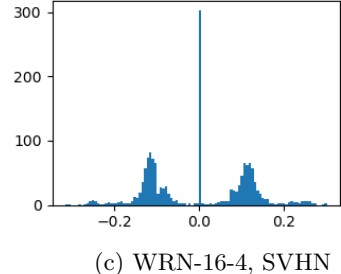

(a) WRN-40-4, CIFAR-10      (b) WRN-40-4, CIFAR-100      (c) WRN-16-4, SVHN

Figure 15: Distribution of slopes $\mathbf{b}_l$ in WRN-40-4 with dataset CIFAR-10 and CIFAR-100 in (a) and (b) respectively; and distribution of slopes $\mathbf{b}_l$ in WRN-16-4 with dataset SVHN in (c).

## A.3 Comparison of the distribution of the slopes of RReLU for WideResNets

In the main manuscript, the distributions of the slope of RReLU are shown for ResNets for CIFAR-10 and CIFAR-100 datasets. The same with WRN-16-4 and WRN-40-4 architectures are shown in Fig. 15. It is clear in the figure that many slopes take values close to zero, indicating that the corresponding filters are not useful. For WideResNets, a great number of filters could be ignored without any degradation in performance. For accuracy and saving in memory and computation, see Table 1 in the main manuscript.

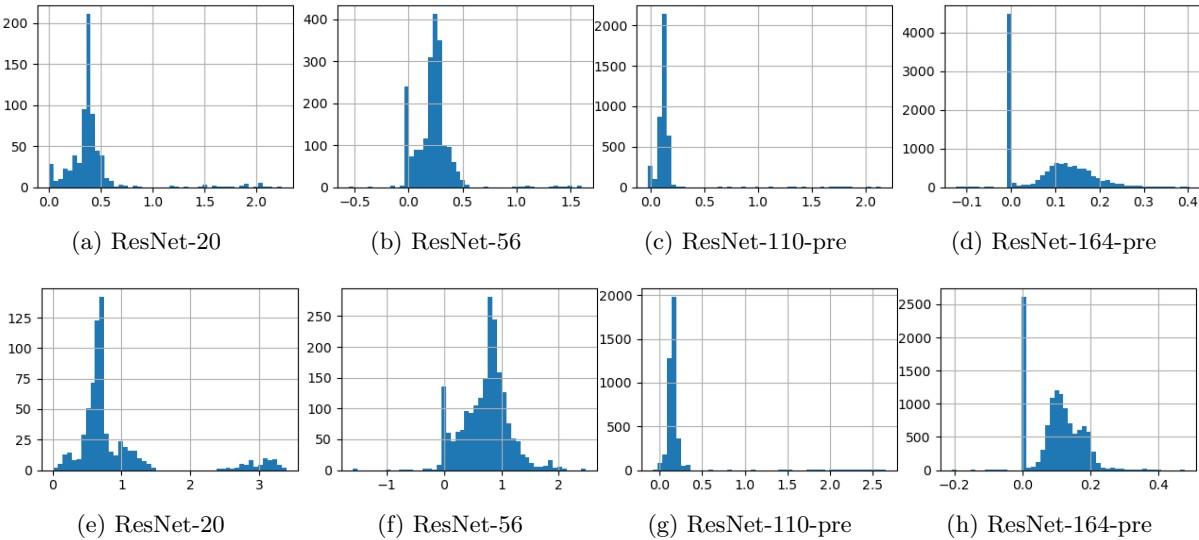

Figure 16: Distribution of BN parameters ($\gamma_l$) with CIFAR-10 (top row) and CIFAR-100 (bottom row).

### A.4 Distribution of the batchnorm paramters

The distribution of batchnorm parameters are presented in Fig. 16 for architectures ResNet-20, ResNet-56, ResNet-110-pre and ResNet-164-pre corresponding to the distribution of the slopes $\mathbf{b}_l$ in Fig. 7 of the main manuscript.

### A.5 Typical values taken by $\gamma_l$ and $\mathbf{b}_l$

In ResNet-164 architecture, there are total of 163 convolution layers and one linear layer at the end before softmax layer. There are a total of 54 Residual blocks out of which we show the scatter plot for a couple of Residual blocks in Fig. 17 which shows the typical values taken by $\gamma_l$ and $\mathbf{b}_l$.

### A.6 Finding the filter-path length

Fig. 13 in the main manuscript shows the filter-path length of WRN-40-4 architecture with RReLU. The metric filter-path length is defined by us, representing the number of filters the features pass through. We get paths of different lengths in ResNets depending on whether we choose the direct or residual connection because features may pass via the residual or direct connections in every ResNet unit Veit et al. (2016). If we consider the direct path, as the number of filters is zero, the value of the filter-path length metric is also zero. Similarly, if we consider the residual path, the features pass via the filters; therefore, the filter-path length is the total number of active filters in that residual unit. As the number of filters reduces at every ResNet unit because of using RReLU, the filter-path length also reduces, indicating that features choose the shortest paths to pass.

### A.7 A discussion on speedup with the sparse models

FLOP reduction serves as an indicative measure of speedup (Gao et al., 2021; Han et al., 2015; Li et al., 2017; Gordon et al., 2018). In our specific computational environment, operating within a 64-bit system, even though the elements of the RReLU slope ($\mathbf{b}_l$) converge to zero, they are still represented as 64-bit real numbers. Consequently, under the current system configuration, any significant speedup is not anticipated. Further, the same model may demonstrate different execution times based on various hardware configurations. For example, in Fig. 15.5.6 of Tu et al. (2022), the peak performance in terms of TFLOPs is different in different deep learning processors. In a single chip also, the performance varies based on the floating

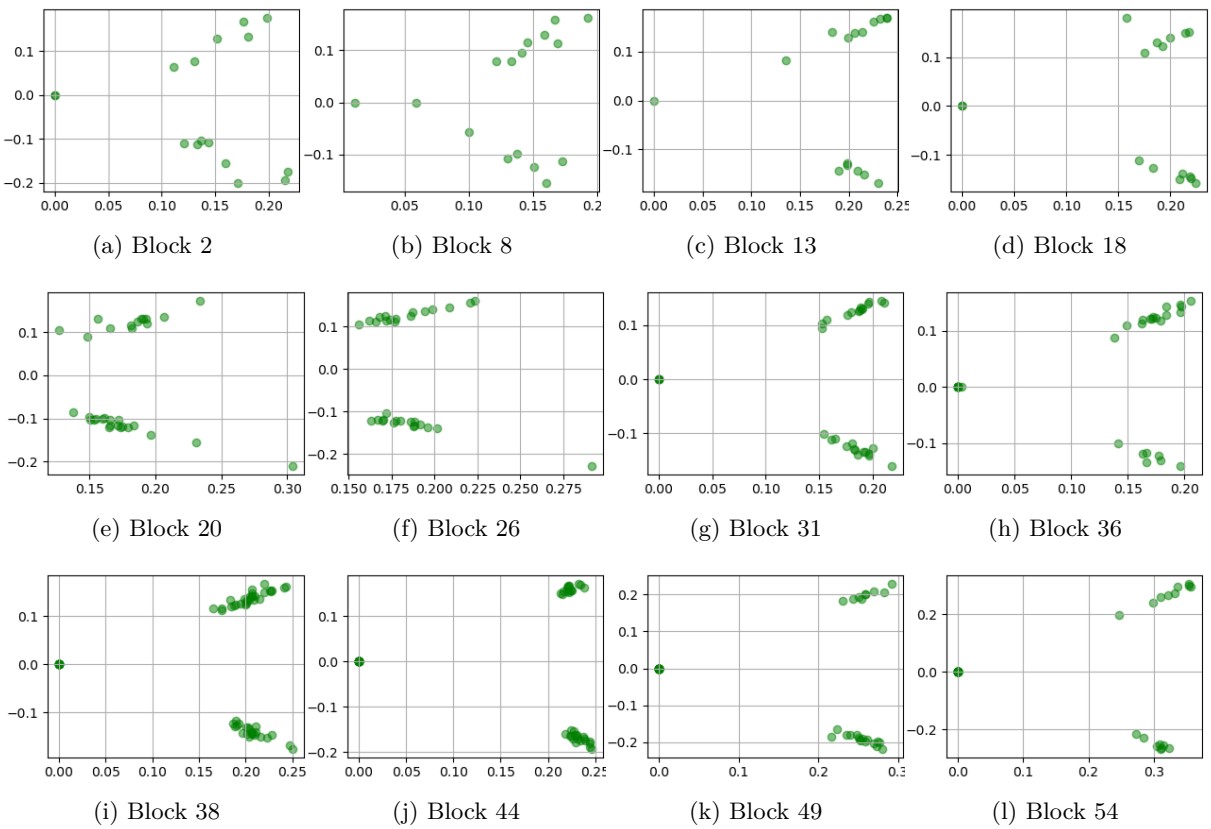

Figure 17: Plot of BN parameters ($\gamma_l$) vs. RReLU slopes ($\mathbf{b}_l$) in different residual blocks of ResNet-164-L1RReLU.

point format (BF16 or FP32). Moreover, the same architecture will have different speedups in different frameworks such as PyTorch, Caffe, Tensorflow, etc (Hadidi et al., 2019). As our focus here is primarily on the algorithmic aspect, hardware/framework considerations are currently not in the scope of this paper.

However, to get an idea about speedup concerning the saving in memory and FLOP reported in the paper, we have done the following. First, we have trained the model with RReLU which makes some of the filters insignificant after training. Here, we have shown the study with a VGG and a ResNet164-pre network. As RReLU can ignore a set of filters and sub-filters in two consecutive layers, for inference we can create a smaller architecture, and overwrite its parameters by the values of the significant filters from the original trained model. The smaller model performs the same as the original model but has a lesser number of filters. Therefore, the inference time of this smaller model will be less. As the inference time is a metric for showing speedup Bianco et al. (2018), therefore we measured the total inference time of both the original model and the smaller model in a GPU device. The below results are averaged over multiple iterations.

| Activation | ReLU (Baseline) | RReLU (Unpruned) | RReLU (Pruned) |
|---|---|---|---|
| FLOP | 1270 Mn | 1270 Mn | 1010 Mn |
| Memory | 20 Mn | 20 Mn | 6.4 Mn |
| Average inference time | 1.06s | 1.03s | 0.62s |
| GPU memory utilization | 81.7% | 81.7% | 77.90% |

Table 10: VGG

In Table 10 and Table 11, we reported the average inference time of VGG and ResNet-164-pre, respectively in a Quadro P2200 5GB RAM GPU while taking average over 200 forward pass. Please note that the

| Activation | ReLU (Baseline) | RReLU (Unpruned) | RReLU (Pruned) |
|---|---|---|---|
| FLOP | 478 Mn | 478 Mn | 307 Mn |
| Memory | 1.7 Mn | 1.7 Mn | 0.92 Mn |
| Average inference time | 3.09s | 3.59s | 2.14s |
| GPU memory utilization | 84.2% | 86.7% | 82.9% |

Table 11: ResNet-164-pre

average inference time of the pruned model using the RReLU slope is lower for both VGG and ResNet-164-pre networks. The GPU utilization is also reported in the last row. We do not have control over the GPU utilization as the GPU uses optimal cores based on batch size and specific GPU configuration. But we have taken a higher batch size to keep them in the same range across all three architectures (i) ReLU (Baseline), (ii) RReLU (Unpruned), and (iii) RReLU (Pruned). For example for VGG, a batch size of 2048 is used whereas for ResNet-164-pre, a batch size of 3000 is used. Keeping the utilization at the same level, we compare the inference time, which shows that RReLU (Pruned) has the minimum inference time therefore it gives the maximum speedup.

## A.8 Distribution of batchnorm scaling parameters and RReLU slopes when both are trainable

We plot the change in distribution of $\gamma_l$ in Fig. 18 and the shift in distribution of $\mathbf{b}_l$ in Fig. 19a, Fig. 19b, and Fig. 19c for the first, second and third layers of every residual blocks, respectively.

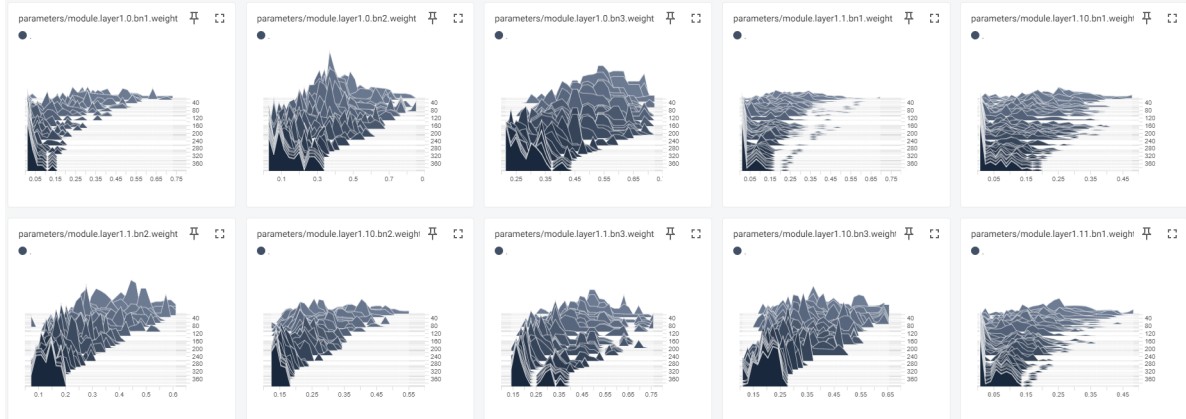

Figure 18: Distribution of $\gamma_l$ for the fourth case where both $\gamma_l$ and $\mathbf{b}_l$ are trainable

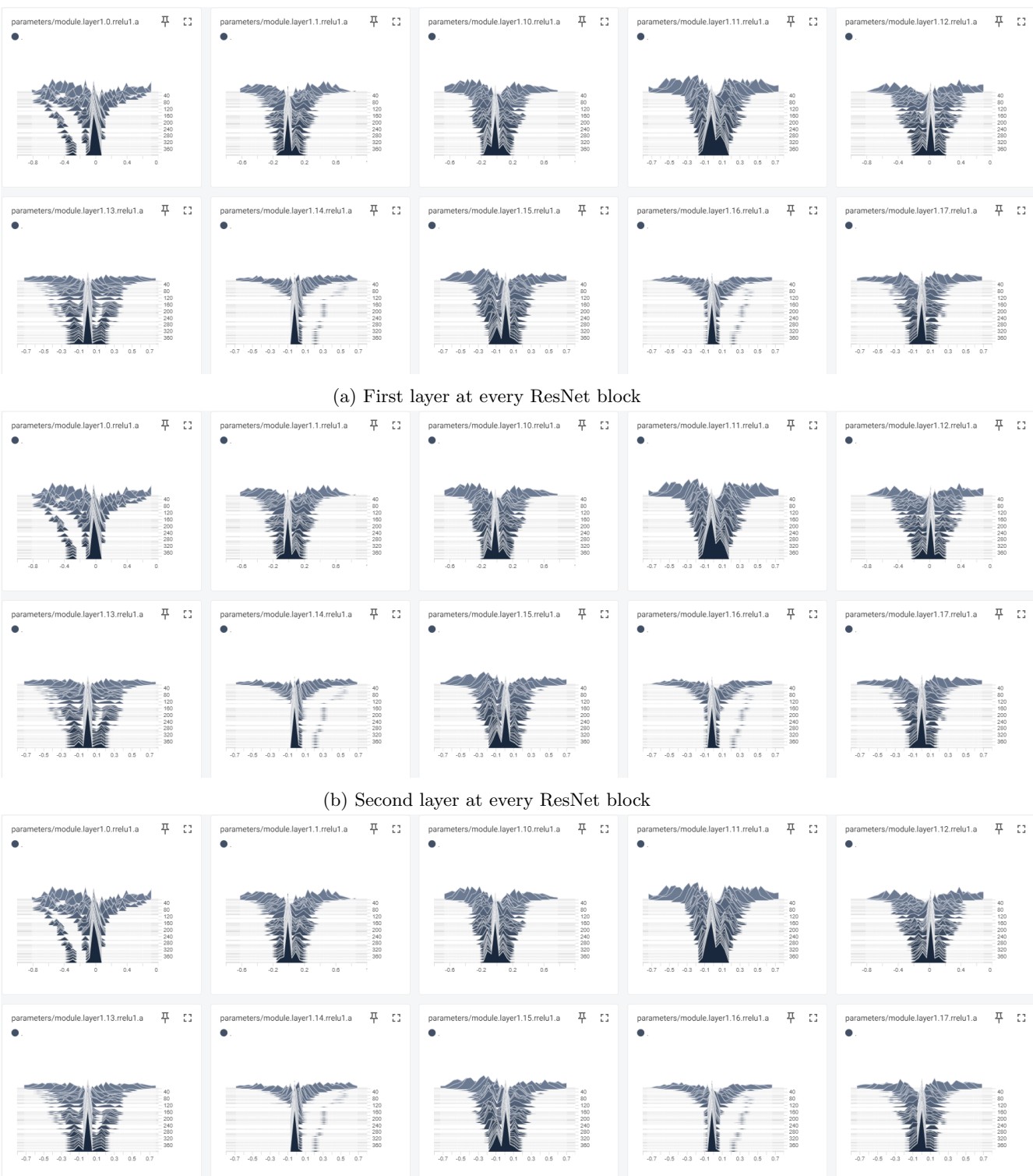

(a) First layer at every ResNet block

(b) Second layer at every ResNet block

(c) Third layer at every ResNet block

Figure 19: Distribution of $\mathbf{b}_l$ for the fourth case where both $\boldsymbol{\gamma}_l$ and $\mathbf{b}_l$ are trainable

