# OpenReview forum: "Rotate the ReLU to Sparsify Deep Networks Implicitly"
_TMLR — Accepted by TMLR_

### Review · Reviewer_7zBv · 2023-10-31

**Summary Of Contributions:**

This work introduces a family of activation functions with trainable parameters, where the magnitude of the multiplier, proportional to the slope of activation, serves as a measure of feature importance. The introduced family comprises several widely used activations - ReLU, GELU. Approach is validated on several computer vision models and benchmarks and compared with L1-BN regularization. In addition to learned sparsity networks parametric activations are shown to be more adversarially robust.

**Audience:**

Yes

**Broader Impact Concerns:**

No need to add Broader Impact Statement

**Claims And Evidence:**

Yes

**Requested Changes:**

Changes

- I would recommend to compare with several stronger baselines to fully appreciate the benefits and contribution of the introduced method.

- The speedups could significantly vary between hardware and details of the environment. Nevertheless, I think this work could benefit from reporting GPU (or CPU) speedups on some configuration.

- ImageNet results are important as proof of the method scalability. The current version doesn’t specify the training recipe (number of training epochs, augmentation/regularization pipeline) used for training of Wide-ResNet and ViT-S/16. The latter model was trained on proprietary JFT data in the original work before finetuning on ImageNet-1k. Does this work apply the same pretraining procedure? Are the sparse models trained with the same compute budget as the dense models?

**Strengths And Weaknesses:**

**Strengths**

- The proposed approach is sound and quite generic. RReLU networks have higher FLOP reduction for the same accuracy compared to L1-BN baseline.

- This work introduces an elaborate initialization scheme for learnable parameters in activations and provides an empirical study on the impact of the initialization and the evolution of trainable PReLU slopes during training. Authors identify the impact of PReLU on batchnorm parameters and corresponding compressibility.

- Networks with parametric activations are noticeably more robust to standard adversarial attacks compared to vanilla ReLU networks.

**Weaknesses**

- Method requires substantially longer training compared to the original dense training recipe to achieve good performance. Whereas such extended training is affordable for CIFAR-10/100, similar procedure for larger datasets, i.e ImageNet-1k becomes rather expensive. Pruned model achieves performance close to the dense model trained for longer, however, for such long training dense model could have reached optimal performance and already entered an overfitting regime. Sparse networks converge slower and benefit more from extended training, as recent work [1] shows.

- The method is missing comparison with several more recent structured pruning methods - Chex [2], EagleEye [3].

- The param and FLOP reduction is provided, but the actual speed-ups are not reported.

- ImageNet-1k training setup is not described in sufficient detail.

---
[1] Kuznedelev, Denis, et al. "Accurate Neural Network Pruning Requires Rethinking Sparse Optimization." arXiv preprint arXiv:2308.02060 (2023).

[2] Z. Hou, M. Qin, F. Sun, X. Ma, K. Yuan, Y. Xu, Y.-K. Chen, R. Jin, Y. Xie, and S.-Y. Kung, “Chex: Channel exploration for cnn model compression,” in Proc. IEEE Conf. Comput. Vis. Pattern Recog., 2022, pp. 12 287–12 298.

[3] B. Li, B. Wu, J. Su, and G. Wang, “Eagleeye: Fast sub-net eval- uation for efficient neural network pruning,” in Proc. Eur. Conf. Comput. Vis. Springer, 2020, pp. 639–654.

---

> ### Author Response · Authors · 2024-03-04
> **Response to Reviewer 7zBv : Part 1**
>
> Thank you for your thoughtful comments and for your patience in awaiting our reply. We have carefully reviewed each of your suggestions and have taken action to address them in the revised version of the paper that we have just submitted. We appreciate your valuable feedback and hope that our revisions satisfactorily address your comments.
>
> ## (Weakness1.a)
> We acknowledge the reviewer’s concern about resource efficiency and the potential cost implications of longer training times, especially for researchers with limited access to computational resources. However, in applications where a sparse model is required to deploy in the resource constrained edge network, a little longer training time is justified. In this context, we would like to clarify that a longer training time required for RReLU is essential for ensuring the convergence of RReLU slopes. However, the extended training time till $1200$ epochs only applies to ResNet-(20/56/110-pre/164-pre) models in Table 1 and not other results with VIT-s16, WRN-50-2 or EfficientNet-b4. Prior works train baseline ResNet-(20/56/110-pre/164-pre) models with ReLU for $200$ epochs with a multi-step lr-scheduler, achieving accuracies listed under "Acc ReLU (standard)" row in Table 1 of the manuscript. But, we observed that using a cosine annealing (CA) lr-scheduler leads to better convergence for both ReLU and RReLU but takes more time to converge, with performance continuing to increase up to $1000$ epochs. For example, under "Acc ReLU (more training)" in Table 1, the baseline accuracy for ResNet-110-pre improves from $93.63\%$ to $95.33\%$. Therefore, to ensure a fair comparison, we decided to use CA LR-scheduler for both ReLU and RReLU, train upto $1200$ epochs so that accuracy saturates well for both of them and models can explore their full potential. One can always stop training early if the old baseline accuracy suffices for the task. To show that even at the end of $400^{th}$ epoch the RReLU provides intrinsic sparsification, we have trained ResNet-164-pre with both ReLU and RReLU for $400$ epochs and reported the parameter and FLOP count below.
> |Architecture| Accuracy |#Param(Mn) | #FLOP (Mn)| Filters ignored (%) |
> |---|---|---|---|---|
> | ReLU | $95.77$| $1.67$ | $489$| - |
> |RReLU|$95.54$ | $1.07$ |$362$ | $31.96$|
> Table: Comparing the sparsity when both the architectures are trained for 400 epochs
>
> Similarly, even though existing works trains WRN-40-4 for $200$ epochs, as we train WRN-40-4 model with RReLU for $400$ epochs, we train the WRN-40-4 with ReLU also for $400$ epochs.
>
> Regarding the reviewer's concern about the extended training for larger datasets like ImageNet, we would like to mention that the number of training epochs for RReLU is comparable to the same for ReLU. For instance,
> 1. ViT-s16 is trained for $300$ epochs, whereas with RReLU, it is trained for $400$ epochs.
> 2. WRN-50-2 architecture with both ReLU and RReLU are trained for $120$ epochs, which are the same as given in the official PyTorch implementation.
> 3. We added more results with the SiLU activation used in EfficientNet for ImageNet. Wheras architecture with SiLU is trained for $400$ epochs, the same with RSiLU is trained for $500$ epochs.
> These results demonstrate that our approach does not require substantially longer training times; the slight increase in training duration is necessary to ensure that the RReLU slopes converge effectively. The results with EfficientNet is added in **Sec. 5.9 Scalable across larger dataset and various architectures**.
>
> ## (Weakness1.b)
> We have cited [1] in Sec. 5 of the main manuscript. The RReLU slope values are initialized with a truncated Gaussian Mixture Model centered at {+1,-1} with variance 3 to prevent quick convergence to zero. Initializing them such higher values requires more time for convergence. The CA lr-scheduler explore the parameter space well. However, it takes more time to converge than multi-step. So when we trained both ReLU and RReLU for $1200$ epochs with CA, the accuracy for both ReLU and RReLU were continuously increasing and saturated by $1200^{th}$ epoch. We did not observe overfitting for the models in Table 1.
>
> [1] Kuznedelev, Denis, et al. "Accurate Neural Network Pruning Requires Rethinking Sparse Optimization." arXiv preprint arXiv:2308.02060 (2023).

---

> ### Author Response · Authors · 2024-03-04
> **Response to Reviewer 7zBv : Part 2**
>
> ## (Weakness2, Q1)
> We thank the reviewer for suggesting a comparison with CHEX [2] and EagleEye [3]. As CHEX is shown to outperform EagleEye, we have shown a comparison with CHEX. The authors in [2] propose a channel exploration strategy to prune and regrow the channels throughout the training process repeatedly. Channel pruning is performed by a column subset selection (CSS) formulation, where, for the regrowing stage, they dynamically re-allocate channels across all the layers under a global channel sparsity constraint. The pruning and regrowing are performed during the training after every $\Delta T = 2$ time step.
>
> In the previous comment, the reviewer rightly pointed out that the model takes more time to train when incorporated with different pruning mechanisms. The CHEX performs channel pruning by Column Subset Selection, which involves finding singular vectors of the network parameters, i.e., the channel matrices of every layer. The SVD is a computationally heavy process [7] [8]; so even though the models are trained for same number of epochs, the pruning and regrowing stage in between the training epochs takes more time. So, we expected a higher training time per epoch for CHEX. To verify this, we ran $10$ epochs of all three methods (i) CHEX, (ii) ReLU, and (iii) RReLU with the same ResNet50 architecture on an NVIDIA A100 GPU. $10$ epochs of training took $315.87$ minutes for CHEX, whereas the same for ReLU and RReLU are  $233.63$ minutes and $276.18$ minutes, respectively. This shows that even though CHEX is trained for the same $250$ epochs, the total training time is higher because of the computationally heavy SVD method for column subset selection.
>
> CHEX considers ResNet50 architecture for the ImageNet classification task and achieves an accuracy of $76.13$% using a GFLOP count of $4.09$. The ResNet50 architecture with RReLU reduces the GFLOP to $3.45$ without using any regularizer. This shows that RReLU can provide sparsity intrinsically. Here, both the architectures with ReLU and RReLU, respectively, are trained for $250$ epochs. We have observed that instead of the usual multi-step learning rate (LR) scheduler, if the cosine annealing LR scheduler is used, then the accuracy improves to $76.28$% with $3.15$ GFLOPs. The improved performance indicates that if trained well with appropriate hyperparameters and proper initialization, the architectures with RReLU can have better accuracy than ReLU with a lesser number of GLOPs within the same number of time steps.
>
> We acknowledge that compared to recent sparsity-inducing methods such as CHEX, RReLU provides less sparsity. However, it's crucial to recognize that the sparsity induced by RReLU is intrinsic. As highlighted in our paper, RReLU is an activation function that inherently promotes sparsity and is not a pruning technique. Consequently, RReLU can be effectively combined with various structured pruning methods to enhance sparsity, as each filter has better representation power with RReLU. To validate this, we have integrated RReLU with CHEX and compared the GFLOP counts for ResNet50 with ReLU and RReLU below. To reach to a comparable performance as ReLU, we have trained the combined CHEX-RReLU architecture for $400$ epochs.
>
> | Methods | Accuracy | GFLOPs|
> |---|---|---|
> |  ReLU (multistep LR) | $76.13$% | $4.09$ |
> |  RReLU (multistep LR) | $75.22$% | $3.45$ |
> |  RReLU (cosine annealing) | $\mathbf{76.28}$% | $3.15$ |
> CHEX w/ ReLU ($s=70$%, cosine annealing) | $76.28$% | $1.35$ |
> CHEX w/ RReLU ($s=70$%, cosine annealing) | $75.48$%  | $\mathbf{1.12}$|
> CHEX w/ ReLU ($s=50$%, cosine annealing) | $77.98$% | $2.49$|
> CHEX w/ RReLU ($s=50$%, cosine annealing) | $77.58$%  | $\mathbf{2.20}$ |
> Table: Comparison between (i) ReLU (ii) RReLU (ii) CHEX, and (iv) RReLU combined with structured pruning method CHEX for ResNet50 on ImageNet dataset.
>
> With a target sparsity of $s=70$%, the baseline (CHEX w/ ReLU) with ResNet50 on ImageNet classification achieved an accuracy of $76.28$% with a GFLOP count of $1.35$ when trained from scratch using the codebase and the hyperparameters the authors have provided in [4]. When CHEX is combined with RReLU (CHEX w/ RRELU), the architecture reduces the computation to $1.12$ GFLOPs, as shown in the above Table, corroborating our claim. Further, with a target sparsity of $s=50$%, CHEX reduces the computation from $4.09$ GFLOPs to $2.20$ GFLOPs, but when CHEX is combined with RReLU, it reduces the computation to $2.20$ GFLOPS. This shows that RReLU improves the sparsity when combined with existing structured pruning techniques. The detailed results are provided in the above table. We have added the experiments regarding CHEX in **Sec. 5.6 RReLU combined with other structural pruning method improves sparsity** of the revised manuscript.
>
> [4] https://github.com/zejiangh/Filter-GaP/tree/main

---

> ### Author Response · Authors · 2024-03-04
> **Response to Reviewer 7zBv : Part 3**
>
> ## (Weakness3, Q2)
> FLOP reduction serves as an indicative measure of speedup [9,10,11,12]. In our 64-bit system, the RReLU slope elements converge to zero but are still represented as 64-bit real numbers. Therefore, we don't expect significant speedup in our current setup.
>
> As the reviewer suggested, we studied more about deploying sparse model in hardware and figured out that Distillers (https://intellabs.github.io/distiller/index.html) are used for prototyping and analyzing compression algorithms for hardware implementation. Even while using distillers, the compression schedulers need to be designed carefully to handle filter/channel level pruning, because with parallel-data dependencies (paths), such as ResNets (skip connections) things become increasingly more complex and require a deeper understanding of the data flow in the model. Further, regarding deploying the ML models in hardware, we've engaged in discussions with our colleagues specializing in the same, and we concur with the reviewer that the same model may demonstrate different execution times based on various hardware configurations. For example, in Fig. 15.5.6 of [13], the peak performance in terms of TFLOPs is different in different deep learning processors. In a single chip, the performance varies based on the floating point format (BF16 or FP32). The same architecture will have different speedup in different framework such as PyTorch, Caffe, Tensorflow, etc [14]. As our focus here is primarily on the algorithmic aspect, hardware/framework considerations are currently not in the scope of this paper.
>
> However, to get an idea about speedup with respect to the saving in memory and FLOP reported in the paper, we have done the following. First, we have trained the model with RReLU which makes some of the filters insignificant after training. In this response, we have shown the study with a VGG and a ResNet164-pre network. As RReLU can ignore a set of filters and sub-filters in two consecutive layers, for inference we can create a smaller architecture, and overwrite its parameters by the values of the significant filters from the original trained model. The smaller model is performs same as the original model but has lesser number of filters. Therefore, the inference time of this smaller model will be less. As the inference time is a metric for showing speedup [5], therefore we measured the total inference time of both the original model and the smaller model in a GPU device. The below results are averaged over multiple iterations.
>
>
> |  Activation & ReLU (Baseline) | RReLU (Unpruned) | RReLU (Pruned) |
> |---|---|---|
> |  FLOP |	$1270$ Mn | $1270$ Mn | $1010$ Mn |
> | Memory & $20$ Mn | $20$ Mn | $6.4$ Mn	 |
> | Average inference time | $1.06$s	 | $1.03$s	 | $0.62$s|
> | GPU memory utilization | $81.7$% |	$81.7$% |	$77.90$% |
> |---|---|---|
> Table: VGG
>
> |  Activation & ReLU (Baseline) | RReLU (Unpruned) | RReLU (Pruned) |
> |---|---|---|
> | FLOP | $478$ Mn | $478$ Mn |	$307$ Mn |
> |Memory | $1.7$ Mn| $1.7$ Mn | $0.92$ Mn |
> | Average inference time | $3.09$s	| $3.59$s	| $2.14$s|
> | GPU memory utilization | $84.2$% | $86.7$%|	$82.9$% |
> |---|---|---|
> Table: ResNet-164-pre
>
> Among the first two tables above, in the first and second table, we reported the average inference time of VGG and ResNet-164-pre, respectively in a Quadro P2200 5GB RAM GPU while taking average over $200$ forward pass. Please note that the average inference time of the pruned model using RReLU slope is lower for both VGG and ResNet-164-pre networks. The GPU utilization is also reported at the last row. We do not have a control over the GPU utilization as the GPU uses optimal cores based on batch size and specific GPU configuration. But we have taken a higher batch size to keep them in the same range across all three architectures (i) ReLU (Baseline), (ii) RReLU (Unpruned) and (iii) RReLU (Pruned). For example for VGG, a batch size of $2048$ is used whereas for ResNet-164-pre, a batch size of $3000$ is used. Keeping the utilization in the same level, we compare the inference time, which shows that RReLU (Pruned) has the minimum inference time therefore it gives the maximum speedup. We have included this study in **Sec. 5.11 A discussion on speedup with the sparse models**.

---

> ### Author Response · Authors · 2024-03-04
> **Response to Reviewer 7zBv : Part 4**
>
> ## (Weakness4, Q3)
> For training WRN-50-2, we closely follow the official implementation by PyTorch for image classification with ImageNet dataset. We do not use pre-trained parameters for initializing the network and train WRN-50-2 from scratch for $120$ epochs for both ReLU and RReLU. We utilized the framework given by [6] to present results with the VIT-s16 model which uses standard data augmentation and no sophisticated regularization techniques such as augmentation or regularization pipeline. It does not use any pre-training procedure. The VIT-s16 model for RReLU is trained for $400$ epochs and has been compared with the results in [6]. We train every model, be it with ReLU or RReLU with the same compute budget. We have provided training details now highlighted in **Sec. 5.9 Scalable across larger dataset and various architectures** of the revised manuscript.
>
> Extra: The FLOP calculations are revised for all the models in the revised manuscript.
>
> We sincerely thank the reviewer for the comments which helped us to improve the quality of our manuscript. We hope the revisions are helpful. Please let us know if there's anything else we can address.
>
>
>
>
> References
>
> [2] Z. Hou, M. Qin, F. Sun, X. Ma, K. Yuan, Y. Xu, Y.-K. Chen, R. Jin, Y. Xie, and S.-Y. Kung, “Chex: Channel exploration for cnn model compression,” in Proc. IEEE Conf. Comput. Vis. Pattern Recog., 2022, pp. 12 287–12 298.
>
> [3] B. Li, B. Wu, J. Su, and G. Wang, “Eagleeye: Fast sub-net eval- uation for efficient neural network pruning,” in Proc. Eur. Conf. Comput. Vis. Springer, 2020, pp. 639–654.
>
> [5] Bianco, Simone, Remi Cadene, Luigi Celona, and Paolo Napoletano. "Benchmark analysis of representative deep neural network architectures." IEEE access 6 (2018): 64270-64277.
>
> [6] Steiner, Andreas, Alexander Kolesnikov, Xiaohua Zhai, Ross Wightman, Jakob Uszkoreit, and Lucas Beyer. "How to train your vit? data, augmentation, and regularization in vision transformers." arXiv preprint arXiv:2106.10270 (2021).
>
> [7] Foster, Blake, Sridhar Mahadevan, and Rui Wang. "A GPU-based approximate SVD algorithm." In Parallel Processing and Applied Mathematics: 9th International Conference, PPAM 2011, Torun, Poland, September 11-14, 2011. Revised Selected Papers, Part I 9, pp. 569-578. Springer Berlin Heidelberg, 2012.
>
> [8] Zhang, Anru, and Dong Xia. "Tensor SVD: Statistical and computational limits." IEEE Transactions on Information Theory 64, no. 11 (2018): 7311-7338.
>
> [9] Gao, Shangqian, Feihu Huang, Weidong Cai, and Heng Huang. "Network pruning via performance maximization." In Proceedings of the IEEE/CVF Conference on Computer Vision and Pattern Recognition, pp. 9270-9280. 2021.
> [10] Han, Song, Jeff Pool, John Tran, and William Dally. "Learning both weights and connections for efficient neural network." Advances in neural information processing systems 28 (2015).
>
> [11] Li, Hao, Asim Kadav, Igor Durdanovic, Hanan Samet, and Hans Peter Graf. "Pruning filters for efficient convnets." arXiv preprint arXiv:1608.08710 (2016).
>
> [12] Gordon, Ariel, Elad Eban, Ofir Nachum, Bo Chen, Hao Wu, Tien-Ju Yang, and Edward Choi. "Morphnet: Fast & simple resource-constrained structure learning of deep networks." In Proceedings of the IEEE conference on computer vision and pattern recognition, pp. 1586-1595. 2018.
>
> [13] Tu, Fengbin, Yiqi Wang, Zihan Wu, Ling Liang, Yufei Ding, Bongjin Kim, Leibo Liu, Shaojun Wei, Yuan Xie, and Shouyi Yin. "A 28nm 29.2 TFLOPS/W BF16 and 36.5 TOPS/W INT8 reconfigurable digital CIM processor with unified FP/INT pipeline and bitwise in-memory booth multiplication for cloud deep learning acceleration." In 2022 IEEE International Solid-State Circuits Conference (ISSCC), vol. 65, pp. 1-3. IEEE, 2022.

---

> > ### Comment · Reviewer_7zBv · 2024-03-11
> >
> > Thank you for detailed response!
> > Clarifications and new experimental results address most of my concerns.
> >
> > While I still think that this work would benefit from reporting speed-ups in a setup close to practical, i.e. with batch_size of order few or few dozens of samples, more standard hardware, I agree, that it is not the main scope of the paper.
> >
> > My suggestion is to move the new experimental results to appendix and add reference to them in the main text.

---

> > > ### Author Response · Authors · 2024-03-12
> > > **Thank you!!**
> > >
> > > Thank you very much for your feedback. As you have requested, we have moved the experiments related to speedup to the Appendix in **Sec. A.7 A discussion on speedup with the sparse models**. We have referred to this experiment in a highlighted paragraph in **Sec. 5.4 Discussion on intrinsic sparsity property of RReLU**. We have uploaded the revised version.
> > >
> > > Please let us know if you have any more comments. Once again, thank you very much for your valuable comments.

---

### Review · Reviewer_82c2 · 2023-11-22

**Summary Of Contributions:**

This paper introduces Rotated ReLU Activation (RReLU), a novel approach to enhance sparsification in vision models. Key highlights include:

* Rotated ReLU Activation (RReLU): This paper propose a Rotated ReLU activation, which involves rotating the ReLU activation function to give it an additional degree of freedom. This rotation, combined with an appropriate initialization scheme, leads to implicit sparsification of the network without the need for regularizers. This method shows significant improvements in the representation capability of network filters and leads to substantial memory and computational savings. For instance, in the WRN-40-4 architecture, RReLU achieved a 63.37% reduction in memory and a 69.7% reduction in FLOP-count, all without compromising performance.
* Compatibility with Other Techniques: RReLU can be used alongside other network compression techniques. An example explored in the study is the combination of RReLU with L1 regularization of batch normalization scaling parameters, which results in enhanced sparsity and accuracy.
* Adversarial Robustness: The study also touches upon the adversarial robustness of networks with RReLU activation. It suggests that networks with RReLU could potentially offer better robustness against adversarial attacks than networks with traditional ReLU activations, due to the inherent properties of RReLU affecting the local Lipschitz constant of the network.

**Audience:**

Yes

**Broader Impact Concerns:**

This research focuses on the compression of existing networks, so there is no need to discuss additional ethical aspects or social issues.

**Claims And Evidence:**

Yes

**Requested Changes:**

1. Several years ago, the results of such experiments might have held significance, but nowadays, datasets like CIFAR-10, MNIST, SVHN, and models like Wide-ResNet are seen as mere toy examples in the context of pruning research. To truly prove the usefulness of this approach, it's essential to demonstrate its effectiveness on more complex, denser networks and to compare it directly with the most recent Structured Pruning research. It's also vital to have a broader range of studies on ImageNet and to explore how these methods perform in applications beyond image classification.
2. It's necessary to find ways to mitigate the disadvantages associated with the extended duration of epochs.
3. I believe there's a need for a 2D graph that effectively shows the relationship between Accuracy and Pruned FLOPS, similar to what's presented in Figure 1 of the EfficientNet paper."

**Strengths And Weaknesses:**

Strength
- The paper introduces a novel method to sparsify vision networks used in image classification. This is achieved during the training phase by incorporating a new trainable parameter into the ReLU activation function. This approach aims to enhance sparsification without necessitating additional post-training or re-training procedures.
- Additionally, the paper investigates benefits beyond network sparsity, particularly focusing on the aspect of adversarial robustness. The aim is to understand how this method contributes to improving the network's resilience against adversarial attacks.

Weakness
- Due to added trainable parameters in ReLU function, this method requires longer training epochs to explore changed loss landscape. The expansion of the CIFAR-10 experiment from 200 to 1200 epochs may not seem problematic, but when an experiment's duration increases by 4 to 10 times, it raises concerns. The extended experiment length makes comparisons more challenging, and the figures provided in this paper aren't persuasive.
- It's unfortunate that the majority of experimental results are focused on small datasets like CIFAR and primarily on highly over-parameterized networks. The experimentation with ImageNet is limited to very large networks suitable for image classification, such as WRN/ViT, while ignoring numerous studies on structured pruning results for ResNet-18/50. Over-parameterized networks are inherently likely to benefit greatly from compression, thus using them as a basis for pruning research may not be the best approach. Also, using ViT for these experiments is questionable due to its over-parameterization.
- It is a letdown that the research overlooks networks like MobileNet and EfficientNet, which have shown superior results in vision post-ResNet.

---

> ### Author Response · Authors · 2024-03-04
> **Response to Reviewer 82c2 : Part 1**
>
> Thank you sincerely for your insightful feedback and for your patience in awaiting our response. We have taken your comments into careful consideration and have made significant revisions to the paper to address each of your concerns. Please find our responses corresponding to each of the weaknesses/questions below.  A revised version has been submitted, and we hope that it adequately addresses your comments.
>
> ## (Weakness1, Q2)
> We would like to humbly clarify that the training time is not $4$ to $10$ times for all the architectures. The extended training time till $1200$ epochs only applies to ResNet-(20/56/110-pre/164-pre) models in Table 1 and not other results with VIT-s16, WRN-50-2 or EfficientNet-b4. For example, WRN-50-2 for ImageNet is trained for the usual $120$ epochs (performance is in Table 4 of the manuscript). The ViT-s16 for ImageNet is trained for $400$ epochs when the baseline is trained for $300$ epochs. Whereas standard EfficientNet is trained for $400$ epochs, with RReLU we trained for both $400$ and $500$ epochs and we observed that an extended training by just $100$ epochs gives a slightly better accuracy indicating better convergence of the RReLU slopes. A longer training time required for RReLU is essential for ensuring the convergence of RReLU slopes. In our manuscript, we have included information on training time for all the experiments for better clarification.
>
> Prior works train baseline ResNet-(20/56/110-pre/164-pre) models with ReLU for $200$ epochs with a multi-step lr-scheduler, achieving accuracies listed under "Acc ReLU (standard)" row in Table 1 of the manuscript. However, we observed that using a cosine annealing (CA) lr-scheduler leads to better convergence for both ReLU and RReLU but takes more time to converge, with performance continuing to increase up to $1000$ epochs. For example, under "Acc ReLU (more training)" in Table 1, the baseline accuracy for ResNet-110-pre improves from $93.63$% to $95.33$%. Therefore, to ensure a fair comparison, we decided to use the CA LR-scheduler for both ReLU and RReLU, train up to $1200$ epochs so that accuracy saturates well for both of them and models can explore their full potential. One can always stop training early if the old baseline accuracy suffices for the task. To show that even at the end of $400^{th}$ epoch the RReLU provides intrinsic sparsification, we have trained ResNet-164-pre with both ReLU and RReLU for $400$ epochs and reported the parameter and FLOP count below.
> |Architecture| Accuracy |#Param(Mn) | #FLOP (Mn)| Filters ignored (%) |
> |---|---|---|---|---|
> | ReLU | $95.77$%| $1.67$ | $489$| - |
> |RReLU|$95.54$% | $1.07$ |$362$ | $31.96$|
> Table: Comparing the sparsity when both the architectures are trained for 400 epochs
>
> Similarly, even though existing works train WRN-40-4 for $200$ epochs, as we train the WRN-40-4 model with RReLU for $400$ epochs, we train the WRN-40-4 with ReLU also for $400$ epochs.

---

> ### Author Response · Authors · 2024-03-04
> **Response to Reviewer 82c2 : Part 2**
>
> ## (Weakness 2)
> We agree that WRN is overparametrized for the classification task with the ImageNet dataset but regarding the vision transformer, we have considered the smallest vision transformer i.e. VIT-s16 whose total number of parameters is $22.14$ Mn. Out of this, the MLP part has $14.2$ Mn parameters where the GELU activation is used. On the other hand, the ResNet-50 model has $25.5$ Mn parameters. Many of the recent works with ImageNet have shown sparsity for ResNet50 [1]. The motivation behind considering VIT was to show that the idea of rotation applies to the recent Transformer type models as well where ReLU is not used. We see that the rotation applies to any ReLU type activation, such as GELU to provide intrinsic sparsity.
>
> RReLU does not give intrinsic sparsity when applied with architectures ResNet18 for the ImageNet dataset. This implies that ResNet18 is a highly under-parametrized network and all the filters in ResNet18 are needed for the classification task. We trained ResNet18 for $120$ epochs after which ReLU gave an accuracy of $70.29$% and RReLU gave $70.54$%. With RReLU none of the filters could be ignored. One interesting thing to note here is that RReLU gives better accuracy as it is not able to give sparsity. This proves that RReLU provides better representation power when sparsity cannot be achieved. We have tested for intrinsic sparsity for ResNet50 as well when RReLU is applied. ResNet50 achieves an accuracy of $76.13$% using a GFLOP count of $4.09$ whereas when the ResNet50 architecture achieves an accuracy of $76.28$% is trained with RReLU and cosine annealing lr-scheduler, the GFLOP could be reduced to $3.15$ without using regularizer as shown in the below Table. Both the networks are trained for $250$ epochs. This shows that RReLU can provide sparsity intrinsically.
>
> |  Methods | Accuracy | GFLOPs |
> |--|--|--|
> | ReLU (multistep LR) | $76.13$% | $4.09$ |
> | RReLU (multistep LR) | $75.22$% | $3.45$|
> | RReLU (cosine annealing) | $\mathbf{76.28}$% | $3.15$|
> Table: Comparison between (i) ReLU and (ii) RReLU for ResNet50 on ImageNet dataset.
> The above results are added as part of the explanation in **Sec. 5.6 RReLU combined with other structural pruning methods improves sparsity** of the revised manuscript.
>
>
> ## (Weakness 3)
> We thank the reviewer for the suggestion to test the proposed method with EfficientNet which has shown superior performance post-ResNet and is famous for efficient computation. EfficientNet uses (i) depthwise separable convolution and (ii) sequential squeeze and excitation in an inverted bottleneck residual block along with intermediate expansion operation and therefore is a better version than MobileNet which has only depthwise separable convolution. So we proceed to compare with only EfficientNet. EfficientNet uses Sigmoid Linear Unit (SiLU) activation. We show the intrinsic sparsity achieved by using the idea of rotation in one of the EfficientNet architectures i.e. EfficientNet-b4. We replace SiLU with rotated SiLU (RSiLU) and train it. The EfficientNet architectures are trained for $400$ epochs. In the below Table, we list the test accuracy, memory, and FLOPs of the baseline EfficientNet-b4 architecture in the second column. The test accuracy of EfficientNet-b4 architecture with RSiLU is reported after training it for $500$.
>
> |Architectures | SiLU (Baseline) | RSiLU ($400$ epochs) | RSiLU ($500$ epochs) |
> |--|--|--|--|
> | Test Accuracy | $82.80$% | $81.42$%| $\mathbf{81.67}$% |
> | FLOP |              $4.2$ Bn | $\mathbf{3.5}$ Bn    | $\mathbf{3.5}$ Bn  |
> | Memory | $19.3$ Mn | $\mathbf{17.2}$ Mn   | $\mathbf{17.2}$ Mn |
> Table: Intrinsic sparsity in EfficientNet architecture
>
> The baseline architecture EfficientNet-b4 with SiLU activation has $19.3$ Mn parameters and $4.2$ Bn FLOPs and is reported to have an accuracy of $82.80$%, using RSiLU provides an intrinsic sparsity to reduce the parameters and FLOP count to $17.2$ Mn and $3.5$ Bn respectively with a comparable accuracy of $81.67$%. We could run only one iteration and we believe that with better hyperparameter tuning the accuracy can be improved. These experiments are added in the revised manuscript and highlighted in **Sec. 5.9 Scalable across larger dataset and various architectures**.

---

> > ### Author Response · Authors · 2024-03-04
> > **Response to Reviewer 82c2 : Part 3**
> >
> > ## (Q1)
> > As both Reviewer 82c2 and Reviewer 7zBv have requested, we have considered the latest structured pruning work CHEX [1] and compared our RReLU against it in terms of sparsity. We have also shown the sparsity when RReLU is combined with CHEX. The reviewer may please refer to the response of (Weakness2, Q1) of Reviewer 7zBv. Further, the results are discussed in detail in **Sec. 5.6 RReLU combined with other structural pruning method improves sparsity** of the revised manuscript.
> >
> > The above results are with the ImageNet dataset and ResNet-50 architecture. We have also performed experiments with EfficientNet-b4 with Imagenet dataset as asked by the reviewer and shown that RReLU can still offer better sparsity even though EfficientNet is already well-known for its computationally efficient structure. We believe that the added set of experiments strengthens our proposed method.
> >
> >
> > ## (Q3)
> > We thank the reviewer for the suggestion. However, due to limited computing power, we were unable to retrain various versions of EfficientNet to produce a 2D graph similar to Figure 1 of the EfficientNet paper. In the above Table regarding EfficientNet, we present the reduced computations for EfficientNet-b4 as an example. EfficientNet-b4 reportedly requires 66 hours to train with eight A100 GPU cards, while using only one A100 GPU card, it took us 22 days to train EfficientNet-b4 architecture with RReLU. Although we did not test different versions of EfficientNet architectures, applying the rotation idea across various architectures like Vision Transformers and WideResNets resulted in reduced memory usage and FLOPs, which strengthens our claim.
> >
> > We sincerely thank the reviewer for the comments which helped us to improve the quality of our manuscript. We hope the revisions are helpful. Please let us know if there's anything else we can address.
> >
> > References
> >
> > [1] Z. Hou, M. Qin, F. Sun, X. Ma, K. Yuan, Y. Xu, Y.-K. Chen, R. Jin, Y. Xie, and S.-Y. Kung, “Chex: Channel exploration for cnn model compression,” in Proc. IEEE Conf. Comput. Vis. Pattern Recog., 2022, pp. 12 287–12 298.

---

> ### Comment · Reviewer_ZHyv · 2024-03-12
>
> I would like to thank the authors for the detailed response. Most of my concerns are resolved. I especially like the added discussion on $\gamma$ and $\beta$. This part would definitely improve the technical merit of this paper.

---

> > ### Author Response · Authors · 2024-03-13
> > **Thank you!!**
> >
> > Thank you very much! We are really glad that you liked the added discussion on $\boldsymbol{\gamma}_l$ and $\mathbf{b}_l$.
> >
> > Please let us know if you have any more concerns that you want us to address. Once again, thank you very much for your valuable comments.

---

### Review · Reviewer_ZHyv · 2024-02-28

**Summary Of Contributions:**

This paper proposes a novel learnable activation function, RReLU, to halp improve the learning ability of the model while inducing implicit sparsity for model compression. The paper claims the RReLU activation can help choosing less filters for features to pass through, so sparsity can be achieved intrinsically without a regularizer. Better adversarial robustness is also observed from the experiments.

**Audience:**

Yes

**Broader Impact Concerns:**

No concerns on the broader impact.

**Claims And Evidence:**

No

**Requested Changes:**

1. Revision should be made to clearly discuss the impact of regularization on RReLU (both L1 and weight decay), to determine if regularization is necessary for inducing sparsity in the RReLU slopes.
2. The impact on longer training time should be clearly discussed, and experiments should be conducted fairly with the same training budget.
3. The theortical reasoning needs strengthen, like how is the RReLU slope interacting with BN, and why exactly is intrinstic sparsity can be induced. Maybe visualizing the optimization curve of some RReLU slop parameter will help.
4.  The correspondence of results in Table 1-3 needs further explaination, on whether they are trained under the same setting.
5. (Minor issue) When refering to equations, it would be the best to have () around the equation number, e.g. Equ. (1). Also, there are a lot of in-line equations in Sec. 4, making the paragraph hard to read. Some reformatting will help.

**Strengths And Weaknesses:**

## Strength
1. The proposed method is simple, clear, and easy to follow.
2. The proposed method is applied on both ReLU and GeLU, and across a wide range of common deep learning models and tasks, showing consistent performance improvements.
3. The proposed method can lead to improvement in sparsity and robustness, which are important properties for deep learning models

## Weakness
1. The major drawback of the proposed method lies in the need of extensively more training epochs. Results in Table 1-3 are achieved with a mixed of training epochs, making it hard to decide if the comparison is fair. Also the long training epochs itself raise a red flag on the proposed method. It should be explained if the long training needed to just let the model converge or is this needed for RReLU to induce sparsity. Results of RReLU shoudl also be reported at smaller number of training epochs corresponding to common design choices in previous methods.
2. The paper mentions "without a regualrizer" as a motivation in the introduction, but regularizers are still used in a large portion of experimental results of the proposed method. Furthermore, even without explicit L1, regualrizers like weight decay can also draw weight values smaller, leading to sparsity in the ReLU slopes. Whether weight decay is used and what the relationship between these regularizers and the final performance is are not well explained.
3. Some reasoning in the paper is not well grounded. For example, in Sec. 2.1 the paper discusses the interaction between RReLU slope b and BN scaling factor gamma. Since the two scalars are multiplied directly, without a regularization the gradient passing on them should be symmetric, so having both trainable parameters seems not to be necessary. Also in Sec. 4, the paper argues a small slope can lead to small Lipschitz, yet a small slope can be followed by a large scaling factor of BN or larger weight values in the layer, not necessarily constraining the Lipschitz

---

> ### Author Response · Authors · 2024-03-09
> **Response to Reviewer ZHyv: Part 1**
>
> Thank you sincerely for your insightful feedback. We have made significant revisions to the paper to address your concerns. A revised version has been submitted, and we hope that it adequately addresses your comments. We are still working on your 3rd comment based on the 3rd point in the weakness and will update you on this shortly.
>
> ## (Weakness1, Q2)
>
> We humbly want to clarify that the extended training time till $1200$ epochs only applies to ResNet-(20/56/110-pre/164-pre) models in Table 1 and not other results with VIT-s16, WRN-50-2 or EfficientNet-b4. For example, WRN-50-2 for ImageNet is trained for the usual $120$ epochs (in Table 5 of the manuscript). The ViT-s16 for ImageNet is trained for $400$ epochs when the baseline is trained for $300$ epochs. Whereas standard EfficientNet is trained for $400$ epochs, with RReLU, we trained for both $400$ and $500$ epochs, and we observed that an extended training by just $100$ epochs gives slightly better accuracy, indicating better convergence of the RReLU slopes and better sparsity. We have included the training time information for all the experiments for better clarification in the manuscript.
>
> Prior works train baseline ResNet-(20/56/110-pre/164-pre) models with ReLU for $200$ epochs with a multi-step lr-scheduler, achieving accuracies listed under "Acc ReLU (standard)" row in Table 1 of the manuscript. But, we observed that using a cosine annealing (CA) lr-scheduler leads to better convergence for both ReLU and RReLU but takes more time to converge, with performance continuing to increase up to $1000$ epochs. For example, under "Acc ReLU (more training)" in Table 1, the baseline accuracy for ResNet-110-pre improves from $93.63\%$ to $95.33\%$. Therefore, to ensure a fair comparison, we decided to use CA LR-scheduler for both ReLU and RReLU, train up to $1200$ epochs so that accuracy saturates well for both of them and models can explore their full potential. One can always stop training early if the old baseline accuracy suffices for the task. Similarly, even though existing works train WRN-40-4 for $200$ epochs, as we train the WRN-40-4 model with RReLU for $400$ epochs, we train the WRN-40-4 with ReLU also for $400$ epochs.
>
> In Table 2, as per the reviewer's request, we have compared all the models after training them for $400$ epochs. We observe that after $400$ epochs, the sparsity of RReLU is the same as L1BN-MT, with the accuracy of RReLU better than L1BN-MT. Please refer to **Table 2 of the revised manuscript** for the modified results. One may inquire as to why L1BN-MT exhibits a higher number of pruned filters but a similar degree of savings compared to RReLU. The explanation lies in the fact that toward deeper layers, RReLU primarily targets larger filters of size $3\times3$, whereas L1BN-MT prunes both $3\times3$ and $1\times1$ filters. In **Fig. 8 of the revised manuscript**, the vertical red lines denote the depths where the convolution operation uses $3\times3$ filters while the rest of the convolution operations use filters with kernel size $1$. It is evident that within the higher depths, RReLU achieves a higher degree of sparsity, resulting in greater savings in memory and FLOP compared to the number of pruned filters. Moreover, in this version, we used a sparsity regularizer of $s=5e-5$, whereas earlier, we used $s=1e-5$. So, the savings in FLOP and the number of parameters increase to a great extent, keeping accuracy in the same range. As shown in Fig. 9 of the revised manuscript, the ResNet-$164$-L1RReLU achieves a higher degree of sparsity within the ranger of higher depth both in terms of bigger ($3\times3$) and smaller ($1\times1$) filters, resulting in higher savings. We request the reviewer to kindly review **Sec. 5.5 RReLU enhances the representation capability of the filters** in the revised manuscript for a detailed explanation.
>
> In Table 2 of the manuscript, the model was trained with RReLU for $400$ epochs. It was observed that the sparsity achieved was lower compared to the model trained for $1200$ epochs (as shown in Table 2 of the previous manuscript). However, both models reached almost the same accuracy by $400$ epochs. This indicates that longer training times contribute to greater sparsity improvements.
>
> ... continued in the next part of response

---

> ### Author Response · Authors · 2024-03-09
> **Response to Reviewer ZHyv: Part 2**
>
> ... Continued (Weakness1, Q2) from previous part...
>
> In Table 3 (Table 4 in revised manuscript), the experiment setup is again for ResNet-20 and ResNet-56 for CIFAR-10 and CIFAR-100, which is the same as in Table 1. However, the focus of this particular study is not to show sparsity but to show that RReLU can extract coarser features. We show that only with trainable RReLU slopes (network weights and biases are fixed after a Kaiming He initialization \cite{he2015delving}) the network is able to achieve some level of learning. This first step was trained for $500$ epochs. Then, in the second step, we fixed the RReLU slopes and trained only the network weights and biases for $700$ epochs. As we perform the study in two steps, we train it in two halves for $500$ and $700$ epochs, respectively.
>
>
> ## (Weakness2, Q1)
>
> Regularization is not necessary to induce sparsity when RReLU is used. All the results in Table 1, Table 3, and Table 5 (in the revised manuscript) do not use a regularizer, which indicates that even without a regularizer, the RReLU can provide sparsity intrinsically. However, RReLU can be used along with existing regularization techniques to increase sparsity even more. This shows the versatility of this activation function. To demonstrate how RReLU can be used with existing regularizers, we have shown an example with ResNet-164 in Table 2. Fig. 9-11 are related to the same experiment. For all other results, whether it is for CIFAR-10/CIFAR-100 on ResNets architectures or the Imagenet dataset on WRN-50-2/VIT-s16/EfficientNet-b4 architectures, we have not used a regularizer. To show that RReLU is applicable along with other regularizing methods, we have shown an experiment with CHEX [1] in Table 5.
>
> Weight decay adds a regularization term to the loss function that penalizes large weights. Weight decay is present during training architectures both with ReLU and RReLU. The weight decay is used to reduce overfitting in machine learning models and, therefore, does not give structural sparsity, which we are interested in.
>
>
> ## (Q4)
> As discussed in the response for Q1, the tables are not trained under same setting and we have discussed the details in the corresponding sections. We have discussed the settings in the response of Weakness1/Q2.
>
> ## (Q5)
> We have followed the Latex settings defined by TMLR. We are not sure if we should change the setting from equation x to Equ. (x). Regarding in-line equations in Sec. 4, we have formatted it appropriately.
>
>
> We sincerely thank the reviewer for the comments which helped us to improve the quality of our manuscript. We hope the revisions are helpful. Please let us know if there's anything else we can address.
>
> References
>
> [1] Z. Hou, M. Qin, F. Sun, X. Ma, K. Yuan, Y. Xu, Y.-K. Chen, R. Jin, Y. Xie, and S.-Y. Kung, “Chex: Channel exploration for cnn model compression,” in Proc. IEEE Conf. Comput. Vis. Pattern Recog., 2022, pp. 12 287–12 298.

---

> > ### Author Response · Authors · 2024-03-12
> > **Response to Reviewer ZHyv: Part 3**
> >
> > We sincerely thank the reviewer for your patience in awaiting our response for (Weakness3, Q3). We have considered your comment carefully and performed an experiment to understand the relation between $\boldsymbol{\gamma}_l$ and $\mathbf{b}_l$. Please find our response below. Another revised version has been submitted to include this study, and we hope that it adequately addresses your comment.
> >
> > ## (Weakness3, Q3)
> >
> > We thank the reviewer for asking this excellent question. We agree with the reviewer that mathematically, the RReLU slopes can be learned by the outer convolution operation. But in the absence of RReLU slopes, just the batchnorm parameters alone cannot render some filters unnecessary to achieve structural sparsity. We suspect the reason is that the batchnorm parameters are usually initialized with a positive value; therefore, they do not explore negative values well. Batchnorm parameters cannot be initialized with a range of both positive and negative values because with negative values, during training, there is a high chance that the output from the batchnorm gets mapped to the negative part of the ReLU slope and the information does not pass to the next layer.
> >
> > To understand the relation between the batchnorm parameters $\boldsymbol{\gamma}_l$ and RReLU slopes $\mathbf{b}_l$, we have done the following experiments. We study the effect of both $\boldsymbol{\gamma}_l$ and $\mathbf{b}_l$ in performance in terms of both accuracy and sparsity. So, we consider four cases:
> > 1. The value of the elements of $\mathbf{b}_l$ is fixed to $1$ which is nothing but ReLU and we train only $\boldsymbol{\gamma}_l$. This is the baseline, i.e., experiment with ReLU.
> > 2. The value of every element of $\boldsymbol{\gamma}_l$ is fixed to the initial value of $0.5$ and train only $\mathbf{b}_l$.
> > 3. We fix elements of both $\boldsymbol{\gamma}_l$ and $\mathbf{b}_l$ to $0.5$ and $1$, respectively.
> > 4. We train both $\boldsymbol{\gamma}_l$ and $\mathbf{b}_l$.
> >
> > All the above models are trained for $400$ epochs without using any regularizer. Below are our observations:
> > 1. The first case is nothing but the baseline where we use a model with ReLU and the $\boldsymbol{\gamma}_l$ is trainable. We achieved an accuracy of $95.79$. In this case, only $755$ elements of $\boldsymbol{\gamma}_l$ are close to zero as we do not use $L_1$ regularization on $\boldsymbol{\gamma}_l$. The distribution of elements of $\boldsymbol{\gamma}_l$ is shown in Fig. 14.a of the revised manuscript. So it is clear that the trainable $\boldsymbol{\gamma}_l$ is not able to induce sparsity much by itself.
> > 2. In the second case, we fix every element of $\boldsymbol{\gamma}_l$ to $0.5$ and learn the RReLU slopes $\mathbf{b}_l$ along with other trainable network parameters. The aim is to understand if $\mathbf{b}_l$ can do the job of $\boldsymbol{\gamma}_l$. We achieve an accuracy of $95.70$, close to the previous baseline. From this, we can see that even if we keep the $\boldsymbol{\gamma}_l$ fixed, the $\mathbf{b}_l$ are still able to do the job of $\boldsymbol{\gamma}_l$. So, it is not necessary to have $\boldsymbol{\gamma}_l$ to reach a similar accuracy as the first case. The distribution of the elements of $\mathbf{b}_l$ are shown in Fig. 14.b. One can observe that the number of slopes close to zero is $1561$, which is higher than the previous case where the number of elements of $\boldsymbol{\gamma}_l$, close to zero, is $755$.
> > 3. In the third case, we fix both $\boldsymbol{\gamma}_l$ and $\mathbf{b}_l$ to see if the network filter weights and biases are enough to reach the same accuracy as the above two cases. We observe that it reaches an accuracy of $95.40$, almost the same as the previous two methods. But please note that as the elements of $\boldsymbol{\gamma}_l$ and $\mathbf{b}_l$ are fixed to values $0.5$ and $1.0$, respectively, the sparsity could not be induced.
> >
> >
> > ... continued in the next part of response

---

> > > ### Author Response · Authors · 2024-03-12
> > > **Response to Reviewer ZHyv: Part 4**
> > >
> > > ... Continued (Weakness3, Q3) from previous part...
> > >
> > > 4. The fourth case has both $\boldsymbol{\gamma}_l$ and $\mathbf{b}_l$ are trainable and the model reaches an accuracy of $95.84$. The distributions of $\boldsymbol{\gamma}_l$ and $\mathbf{b}_l$ are shown in Fig. 14.c and Fig. 14.d of the revised manuscript, respectively. When both $\boldsymbol{\gamma}_l$ and $\mathbf{b}_l$ are trained together, the sparsity is more. We see that $2354$ slopes are near zero. But a slightly lesser number of $\boldsymbol{\gamma}_l$ ($2282$) are close to zero because when the elements of $\mathbf{b}_l$ do not take values close to zero, the elements of $\boldsymbol{\gamma}_l$ take a wide range of values. This is more prominent when the network is trained for more epochs. Comparing Fig. 14.d with Fig. 14.b, one can observe that with trainable $\boldsymbol{\gamma}_l$ and $\mathbf{b}_l$, the sparsity is more than only trainable $\mathbf{b}_l$ even though $\mathbf{b}_l$ can do the job of $\boldsymbol{\gamma}_l$ for achieving good accuracy. Comparing Fig. 14.c with Fig. 14.a shows that with trainable $\boldsymbol{\gamma}_l$ and $\mathbf{b}_l$, the number of $\boldsymbol{\gamma}_l$ close to zero also increases. Further, the sparsity increases if the model is trained for more training epochs as more elements of $\mathbf{b}_l$ go towards zero. In the previous version, the sparsity was higher after training for $1200$ epochs.
> > >
> > > As we could not attach the plots, we request the reviewer to kindly refer to **Sec. 5.11 A study on the sparsification capability of batchnorm scaling parameters versus RReLU slopes** for the plots and the above explanation is added too in Sec 5.11. Towards the end of this section, we have also shown how the distributions change during optimization for some of the $\boldsymbol{\gamma}_l$ in Fig. 18 of the Appendix. Fig. 19.a, Fig. 19.b, and Fig. 19.c in the Appendix show the distribution of $\mathbf{b}_l$ in the first, second, and third layers, respectively, of every residual block.
> > >
> > > We sincerely thank the reviewer for this comment which helped us to improve the quality of our manuscript because of the added experiment to understand the relation between $\boldsymbol{\gamma}_l$ and $\mathbf{b}_l$. Please let us know if there's anything else we can address.

---

### Decision · Action_Editor_AwPk · 2024-04-25

**Recommendation:** Accept as is

**Comment:**

The paper proposes an interesting way to encourage sparsity. The authors used a wide variety of datasets and architectures. The scale and the relevance of the experiments could be improved in the next iteration of this work.

**Audience:**

The study is on the effect of activation function modifications to the sparsity of neural networks, so the relevant audience should include a large part of deep learning researchers.

**Claims And Evidence:**

Authors study the sparsification effect of rotating the relu of neural networks. The experiments include a variety of small datasets and ImageNet, as well as a wide variety of architectures including fully connected neural networks, convolutional networks, and transformers.